# Rabphilin 3A binds the N-peptide of SNAP-25 to promote SNARE complex assembly in exocytosis

Tianzhi Li, Qiqi Cheng, Shen Wang, Cong Ma*

Key Laboratory of Molecular Biophysics of the Ministry of Education, College of Life Science and Technology, Huazhong University of Science and Technology, Wuhan, China

**\*For correspondence:**
cong.ma@hust.edu.cn

**Competing interest:** The authors declare that no competing interests exist.

**Abstract** Exocytosis of secretory vesicles requires the soluble N-ethylmaleimide-sensitive factor attachment protein receptor (SNARE) proteins and small GTPase Rabs. As a Rab3/Rab27 effector protein on secretory vesicles, Rabphilin 3A was implicated to interact with SNAP-25 to regulate vesicle exocytosis in neurons and neuroendocrine cells, yet the underlying mechanism remains unclear. In this study, we have characterized the physiologically relevant binding sites between Rabphilin 3A and SNAP-25. We found that an intramolecular interplay between the N-terminal Rab-binding domain and C-terminal $C_2AB$ domain enables Rabphilin 3A to strongly bind the SNAP-25 N-peptide region via its $C_2B$ bottom α-helix. Disruption of this interaction significantly impaired docking and fusion of vesicles with the plasma membrane in rat PC12 cells. In addition, we found that this interaction allows Rabphilin 3A to accelerate SNARE complex assembly. Furthermore, we revealed that this interaction accelerates SNARE complex assembly via inducing a conformational switch from random coils to α-helical structure in the SNAP-25 SNARE motif. Altogether, our data suggest that the promotion of SNARE complex assembly by binding the $C_2B$ bottom α-helix of Rabphilin 3A to the N-peptide of SNAP-25 underlies a pre-fusion function of Rabphilin 3A in vesicle exocytosis.

## Editor's evaluation

This fundamental work is of interest to cell biologists and neuroscientists working on the molecular event underlying synaptic vesicle fusion. The authors obtain an improved and compelling description of the interaction of rabphilin with the SNARE complex protein SNAP-25. They provide a novel hypothesis of rabphilin function and how its interaction with SNAP-25 may help in the assembly of SNARE complexes.

## Introduction

Secretion of neurotransmitters and hormones mediated by $Ca^{2+}$-regulated exocytosis in neurons and neuroendocrine cells is achieved by a series of intracellular membrane trafficking steps, including recruitment, docking, priming, and fusion of secretory vesicles with the plasma membrane (*Rettig and Neher, 2002*; *Südhof and Rizo, 2011*; *Jahn and Fasshauer, 2012*). The core machinery governing the process involves the SNARE proteins synaptobrevin-2 (Syb2) on secretory vesicles, and syntaxin-1 (Syx1) and SNAP-25 (SN25) on the plasma membrane, which form the four helical bundle structure called the SNARE complex to bring the two membranes into close proximity and eventually drive membrane fusion (*Jahn and Scheller, 2006*; *Sutton et al., 1998*; *Weber et al., 1998*). Small GTPase Rabs are essential regulators in the secretory pathway, which have specific role in defining the identity

of subcellular membranes (*Stenmark, 2009*). Rab3A and Rab27A are abundantly expressed in neurons and neuroendocrine cells and associate with the membrane of secretory vesicles in a GTP-bound form to regulate vesicle exocytosis (*Takai et al., 1996*; *Geppert and Südhof, 1998*; *Fukuda, 2005*). Rabphilin 3A (Rph3A) was originally identified as a GTP-Rab3A-binding protein and found to regulate SNARE-dependent exocytosis (*Shirataki et al., 1992*; *Tsuboi and Fukuda, 2005*; *Deák et al., 2006*). Recent studies have found that loss of Rph3A is associated with Alzheimer's and Huntington's disease, suggesting a crucial role of Rph3A in synaptic function (*Smith et al., 2005*; *Smith et al., 2007*; *Tan et al., 2014*).

Rph3A contains an N-terminal Rab-binding domain (RBD) that associates with secretory vesicles via binding to GTP-bound Rab3A or Rab27A (*Li et al., 1994*; *Mizoguchi et al., 1994*; *Fukuda et al., 2004*), a proline-rich linker (PRL) region bearing multiple phosphorylation sites (*Foletti et al., 2001*), and two C-terminal $C_2$ domains (termed $C_2A$ and $C_2B$, respectively) that interact with phospholipids and the SNAREs (*Yamaguchi et al., 1993*; *Chung et al., 1998*; *Tsuboi and Fukuda, 2005*; *Figure 1A*). In particular, increasing evidence showed that the $C_2B$ domain of Rph3A binds SN25 to regulate exocytosis of dense-core vesicles (DCVs) in PC12 cells and fine-tune re-priming of synaptic vesicles in hippocampal neurons (*Tsuboi and Fukuda, 2005*; *Deák et al., 2006*), but the underlying mechanism was unclear. It was previously reported that Rph3A binds SN25 via the β3-β4 polybasic region of the $C_2B$ domain (*Tsuboi et al., 2007*), while later structural results identified that Rph3A interacts with SN25 via the α-helix at the bottom face of the $C_2B$ domain (*Deák et al., 2006*; *Ferrer-Orta et al., 2017*). In addition, according to the crystal structure of the Rph3A-$C_2B$–SN25 complex, three distinct sites in SN25 were found to participate in binding to Rph3A-$C_2B$, two of which located in the middle region of the SNARE motif of SN25, and one resided at the N-terminus of SN25 (*Ferrer-Orta et al., 2017*). However, it remains elusive which binding mode of the Rph3A-$C_2B$–SN25 interaction is physiologically relevant. Importantly, little is known how this interaction regulates vesicle exocytosis.

In this study, we have systematically characterized the interaction between Rph3A and SN25 using full-length form of Rph3A. We found that the N-peptide of SN25 (residues 1–10) mediates the Rph3A–SN25 interaction in a manner that requires an intramolecular interplay between the N-terminal RBD domain and C-terminal $C_2AB$ domain of Rph3A. In addition, we identified that the bottom α-helix rather than the β3-β4 polybasic region in the $C_2B$ domain of Rph3A mediates SN25 interaction. Deletion of the N-peptide of SN25 or disruption of the $C_2B$ bottom α-helix of Rph3A significantly reduced the Rph3A–SN25 interaction and impaired docking and fusion of DCVs with the plasma membrane in PC12 cells, revealing the physiological relevance of such Rph3A–SN25 interaction. Furthermore, we found a stimulatory role of Rph3A in SNARE complex assembly dependent on the Rph3A–SN25 interaction. This stimulatory effect of Rph3A arises because Rph3A induces a conformational change of the SN25 SNARE motif from random coils to α-helical structure.

## Results

### An intramolecular interplay of Rph3A FL contributes to SN25 interaction

Given the difficulties in obtaining full-length Rph3A (Rph3A FL, residues 1–681) proteins in vitro, most previous studies alternatively used various Rph3A fragments to map the potential binding sites between Rph3A and SN25 (*Deák et al., 2006*; *Ferrer-Orta et al., 2017*). Here, we have successfully obtained Rph3A FL with desirable yield in *Escherichia coli* by using a pEXP5-NT/TOPO expression system. Rph3A FL displayed a strong tendency to form monomer in solution, as detected by size-exclusion chromatograph and analytical ultracentrifugation (AUC) (*Figure 1—figure supplement 1*).

We first examined the binding between Rph3A and SN25 (full length, 1–206) by using GST pull-down experiments. We have designed and obtained a series of truncations that selectively comprise one or several domains of Rph3A, that is, Rph3A (1–281), Rph3A (182–681), Rph3A (282–681), and Rph3A (372–681) (*Figure 1A*). As controls, GST alone displayed no binding to Rph3A FL and the truncations. In contrast, GST-SN25 showed significant interaction with Rph3A (182–681), Rph3A (282–681), and Rph3A (372–681), but failed to bind Rph3A (1–281) (*Figure 1B and C*). Consistent with previous observations, these data indicate that the C-terminal $C_2AB$ of Rph3A contributes predominantly to SN25 interaction. Intriguingly, GST-SN25 showed much more stronger binding capacity to Rph3A FL than to the above truncations (*Figure 1B and C*), and titration of Rph3A FL to GST-SN25 yielded a Kd

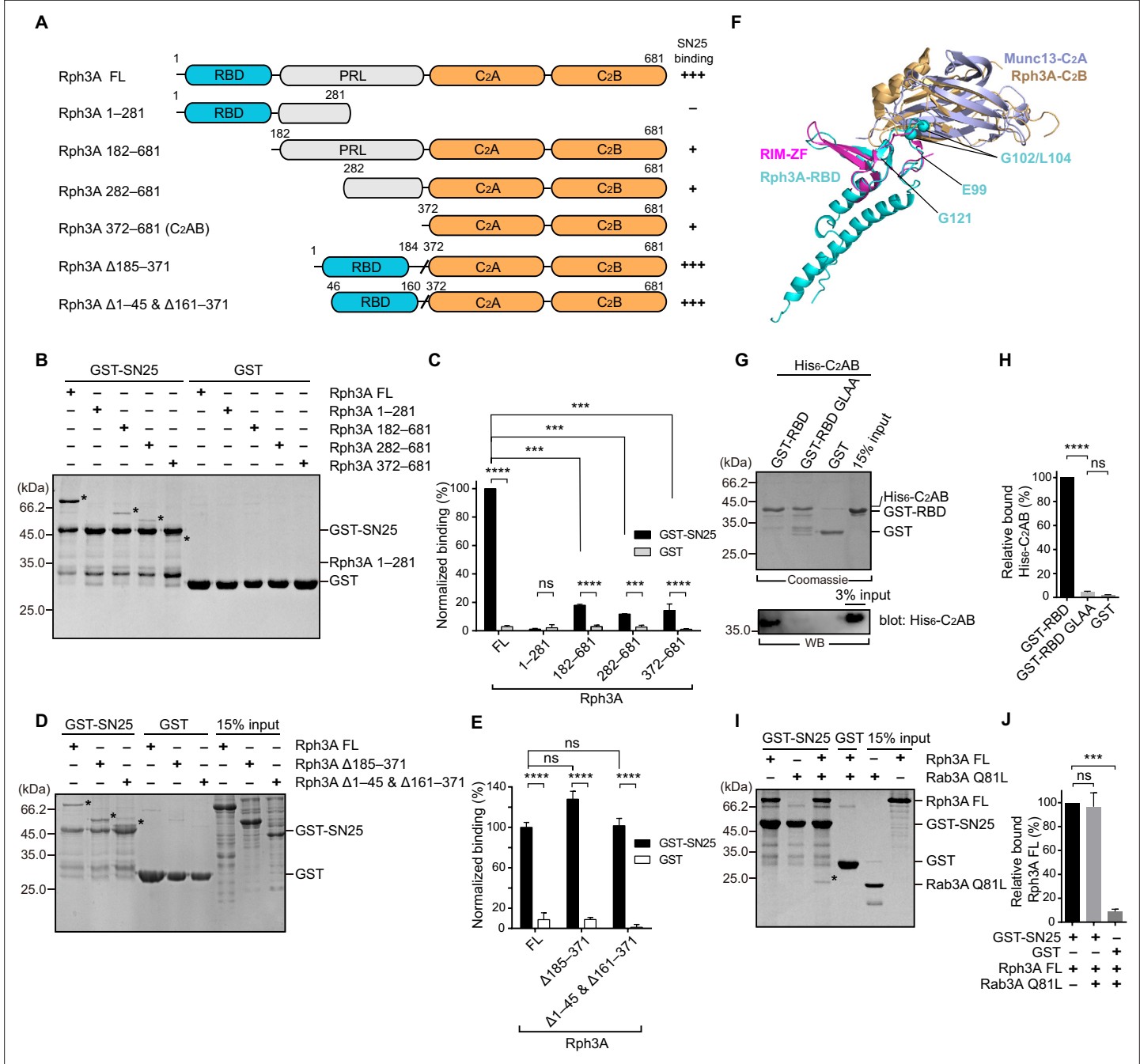

**Figure 1.** The intramolecular interplay of Rph3A enables strong binding to SN25. (**A**) Schematic diagram showing domain organization and variant fragments of Rph3A. The relative SN25 binding affinity (–, +, +++) derived from binding experiments is indicated on the right. (**B, C**) Binding of Rph3A FL or the fragments to GST-SN25 measured by GST pull-down assay (**B**) and quantification of the Rph3A binding (**C**). Asterisks in (**B**) show bands of bound Rph3A FL or fragment proteins. (**D, E**) Binding of Rph3A FL or deletion mutations ($\Delta185–371$, $\Delta1–45$ & $\Delta161–371$) to GST-SN25 measured by GST pull-down assay (**D**) and quantification of the Rph3A binding (**E**). Asterisks in (**D**) show bands of bound Rph3A FL or fragment proteins. (**F**) Structure alignment of Rph3A with RIM-ZF–Munc13-$C_2$A complex (PDB entry: 2CJS). The Rph3A-RBD (PDB entry: 1ZBD) and $C_2$B (PDB entry: 3RPB) domain were aligned and superposed with RIM-ZF, Munc13-$C_2$A, respectively. The G102/L104 site was mapped to Rph3A-RBD structure. (**G, H**) Binding of His$_6$-tagged Rph3A-$C_2$AB to GST-Rph3A-RBD (40–170) or its G102A/L104A mutant (GLAA) measured by GST pull-down assay (**G**) and quantification of the bound His$_6$-$C_2$AB (**H**). The top panel shows a Coomassie blue stained gel of the GST-proteins to illustrate that similar amounts of protein were employed. Bound His$_6$-$C_2$AB proteins were analyzed by immunoblotting with anti-His$_6$ antibody (bottom). (**I, J**) Binding of Rph3A FL to the GST-SN25 in the presence of Rab3A Q81L measured by GST pull-down assay (**I**) and quantification of the bound Rph3A (**J**). Asterisk in (**I**) shows the band of bound Rab3A Q81L. Data are processed by ImageJ (NIH) and presented as the mean ± SEM (n=3), technical replicates. Statistical significance and p values were determined by one-way analysis of variance (ANOVA). ***p<0.001; ****p<0.0001; ns, not significant.

*Figure 1 continued on next page*

*Figure 1 continued*

The online version of this article includes the following source data and figure supplement(s) for figure 1:

**Source data 1.** Uncropped SDS-gel shown in *Figure 1B*.

**Source data 2.** Excel file with data used to make *Figure 1C, E, H and J*.

**Source data 3.** Uncropped SDS-gel shown in *Figure 1D*.

**Source data 4.** Uncropped SDS-gel and Western blot shown in *Figure 1G*.

**Source data 5.** Uncropped SDS-gel shown in *Figure 1I*.

**Figure supplement 1.** Size-exclusion chromatograph (SEC) and analytical ultracentrifugation (AUC) analysis of Rph3A FL protein.

**Figure supplement 1—source data 1.** Uncropped SDS-gel shown in *Figure 1—figure supplement 1B*.

**Figure supplement 2.** Binding Kd between Rph3A FL and GST-SN25.

**Figure supplement 2—source data 1.** Uncropped SDS-gel shown in *Figure 1—figure supplement 2A*.

**Figure supplement 2—source data 2.** Excel file with data used to make *Figure 1—figure supplement 2B*.

**Figure supplement 3.** The effect of G102A/L104A mutant on Rph3A–SN25 interaction.

**Figure supplement 3—source data 1.** Uncropped SDS-gel shown in *Figure 1—figure supplement 2A*.

**Figure supplement 3—source data 2.** Excel file with data used to make *Figure 1—figure supplement 2B*.

---

of 3.293±0.4787 µM (*Figure 1—figure supplement 2*). As Rph3A (1–281) did not bind SN25, the only explanation is that the N-terminal region of Rph3A cooperates with the C-terminal $C_2AB$ to enhance SN25 binding.

To determine which part in the N-terminal region of Rph3A is required for the enhanced SN25 binding, we generated two Rph3A variants, one selectively lacked the middle PRL, that is, Rph3A (Δ185–371), the other additionally removed the N- and C-terminal residues upstream and downstream of the RBD, that is, Rph3A (Δ1–45 and Δ161–371) (*Figure 1A*). Both variants bound to GST-SN25 as effectively as Rph3A FL (*Figure 1D and E*), revealing that the RBD is important for the enhanced SN25 binding. As Rph3A FL prefers a monomeric state in solution and the RBD alone displayed no detectable binding to GST-SN25 as shown in *Figure 1B and C*, we suspect that an intramolecular interplay between the RBD and the $C_2AB$ may enhance Rph3A–SN25 binding. Indeed, we observed that GST-tagged Rph3A-RBD (40–170) but not GST alone bound to $His_6$-tagged Rph3A-$C_2AB$ (372–681), as detected by pull-down assay combined with immunoblotting (*Figure 1G and H*). Next, we sought to explore the binding sites in the RBD that mediate binding to the $C_2AB$. We used the crystal structure of the zinc-finger (ZF) of RIM-1 bound to the $C_2A$ domain of Munc13-1 as a guide (*Lu et al., 2006*), as Rph3A-RBD and Rph3A-$C_2AB$ similarly contain ZF sequence and $C_2$-domain sequence, respectively. Superimposition of the structures of Rph3A-RBD and Rph3A-$C_2B$ with the structure of the RIM-ZF–Munc13-$C_2A$ complex points to a region (residues 99–121) in the RBD (*Figure 1F*). Upon screening, we found that mutation of G102/L104 (G102A/L104A, referred to as GLAA) abolished the binding of Rph3A-RBD to Rph3A-$C_2AB$, as detected by immunoblotting (*Figure 1G and H*). To confirm this, we also introduced the GLAA mutation to Rph3A (Δ185–371) and found that it severely impaired GST-SN25 binding (*Figure 1—figure supplement 3A,B*). Altogether, these data suggest a new binding mode between Rph3A and SN25 which relies on an intramolecular interplay of the RBD with $C_2AB$ in Rph3A.

The RBD of Rph3A can bind Rab3A (Q81L, GTP-bound form) as well (*Stahl et al., 1996*; *Ostermeier and Brunger, 1999*). Based on the crystal structure of the Rph3A-RBD–Rab3A complex (*Ostermeier and Brunger, 1999*), the binding sites for Rab3A position away from residues G102/L104 in the RBD (*Figure 1—figure supplement 3C*), suggesting that the intramolecular interplay of the RBD with $C_2AB$ might be compatible with Rab3A interaction. Indeed, pull-down experiments showed that the involvement of Rab3A did not influence the strong interaction between GST-SN25 and Rph3A FL mediated by the intramolecular interplay of the RBD with $C_2AB$ (*Figure 1I and J*). Instead, Rab3A showed significant binding to the Rph3A FL–SN25 complex (*Figure 1I and J*), implying that Rph3A is able to serve as a connector to bridge vesicle-bound Rab3A and plasma membrane-bound SN25.

## The N-peptide of SN25 mediates Rph3A FL interaction

SN25 contains two SNARE motifs, SNARE motif 1 and 2, connected by a linker region, namely SN1, SN2, and LR, respectively. SN25 also comprises an N-terminal peptide preceding SN1 (*Figure 2A*).

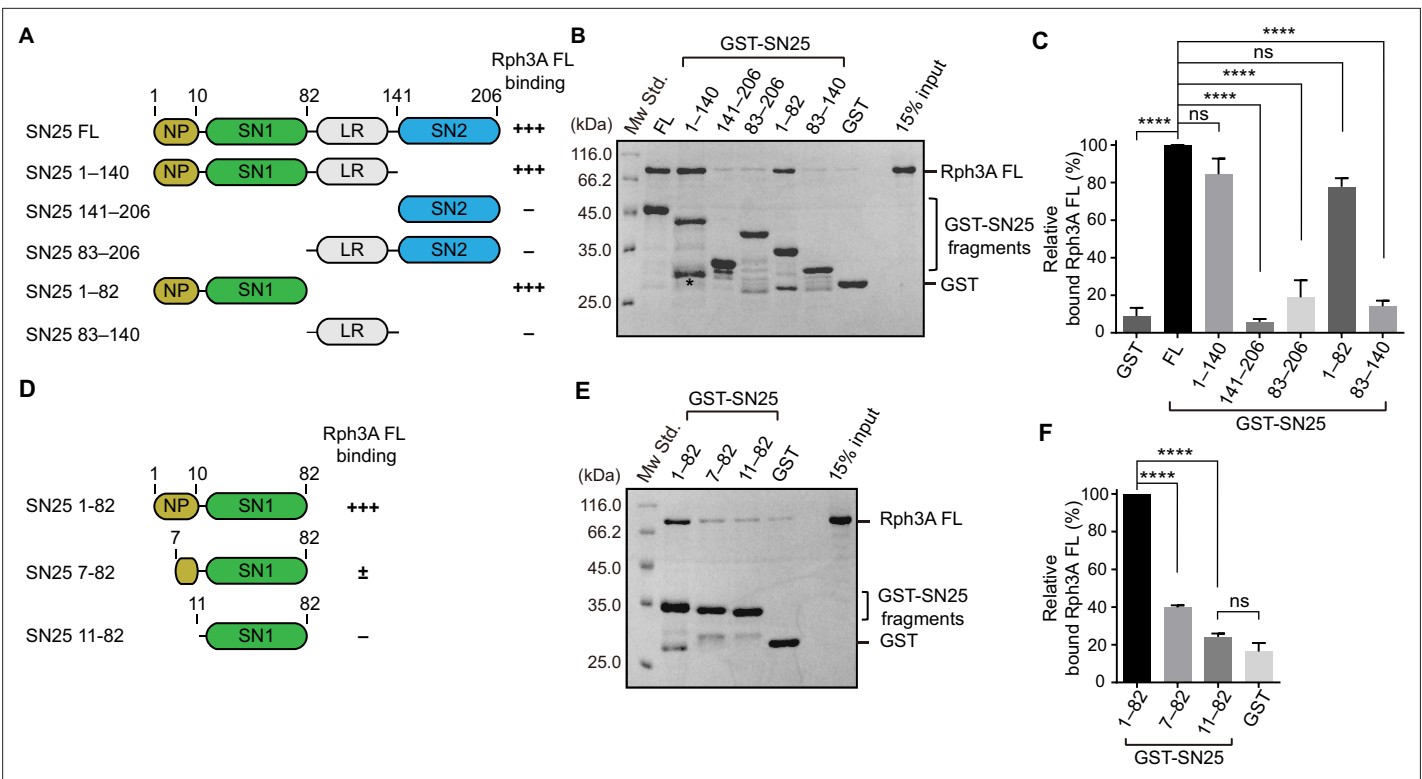

**Figure 2.** Characterization of the Rph3A-binding sites in SN25. (**A**) Schematic diagram showing domain organization and variant fragments of SN25. NP, N-peptide of SN25 (residues, 1–10); SN1, SNARE motif 1; LR, linker region; SN2, SNARE motif 2. The relative Rph3A FL binding affinity (–, +++) derived from binding experiments is indicated on the right. (**B, C**) Binding of Rph3A FL to GST-SN25 FL or fragments measured by GST pull-down assay (**B**) and quantification of the bound Rph3A FL (**C**). Asterisk in (**B**) shows the band of degraded GST-SN25 (1–140) proteins. (**D**) Schematic diagram showing N-peptide deletion mutants of SN25 (1–82). The relative Rph3A FL binding affinity (–, ±, +++) derived from binding experiments is indicated on the right. (**E, F**) Binding of Rph3A FL to GST-SN25 (1–82) or N-peptide deletions (7–82 and 11–82) measured by GST pull-down assay (**E**) and quantification of the bound Rph3A FL (**F**). Data are processed by ImageJ (NIH) and presented as the mean± SEM (n=3), technical replicates. Statistical significance and p values were determined by one-way analysis of variance (ANOVA). ****p<0.0001; ns, not significant.

The online version of this article includes the following source data and figure supplement(s) for figure 2:

**Source data 1.** Uncropped SDS-gel shown in *Figure 2B*.

**Source data 2.** Excel file with data used to make *Figure 2C and F*.

**Source data 3.** Uncropped SDS-gel shown in *Figure 2E*.

**Figure supplement 1.** Effect of middle region mutants in SN1 on Rph3A interaction.

**Figure supplement 1—source data 1.** Uncropped SDS-gel shown in *Figure 2—figure supplement 1A*.

**Figure supplement 1—source data 2.** Excel file with data used to make *Figure 2—figure supplement 1B*.

**Figure supplement 2.** Sequences alignment of SN25 using Clustal Omega Program and Escript 3.0.

**Figure supplement 2—source data 1.** Uncropped SDS-gel shown in *Figure 2—figure supplement 2A*.

**Figure supplement 2—source data 2.** Excel file with data used to make *Figure 2—figure supplement 2B*.

**Figure supplement 3.** The role of conserved acidic residues in N-peptide of SN25 in Rph3A binding.

**Figure supplement 4.** Lipid sedimentation assay to measure the binding of Rph3A–SN25 in the presence of negative membranes.

**Figure supplement 4—source data 1.** Uncropped SDS-gel and statistic data shown in *Figure 2—figure supplement 4*.

Previous structural analysis using the C$_2$B fragment of Rph3A showed that both the middle part of SN1 and the N-peptide of SN25 are involved in Rph3A-C$_2$B interaction (*Ferrer-Orta et al., 2017*). According to the binding mode of Rph3A with SN25 as described above (*Figure 1*), it is necessary to pinpoint the binding sites on SN25 by using Rph3A FL rather than the C$_2$B fragment. To this aim, we designed and obtained a series of SN25 fragments that selectively contains multiple domains, for example, SN25 (1–82), SN25 (1–140), SN25 (141–206), SN25 (83–206), and SN25 (83–140), and

compared their binding abilities with Rph3A FL (*Figure 2A*). Among these fragments, SN25 (1–82) and SN25 (1–140) significantly bound to Rph3A FL, as effectively as SN25 FL (*Figure 2B and C*). In contrast, GST alone and the other SN25 fragments failed to bind Rph3A FL (*Figure 2B and C*). Hence, these data suggest that Rph3A FL binds to the N-terminal region of SN25 comprising the SN1 and the N-peptide.

To further identify the sequence of SN25 responsible for Rph3A FL interaction, we generated two truncations on SN25 (1–82), that is, SN25 (7–82) and SN25 (11–82), with the N-peptide residues differentially deleted (*Figure 2D*). To our surprise, in contrast to the strong interaction of GST-SN25 (1–82) with Rph3A FL, GST-SN25 (7–82) strongly impaired Rph3A FL binding, and GST-SN25 (11–82) failed to bind Rph3A FL (*Figure 2E and F*), showing that the N-peptide sequence ([1]MAEDADMRNE[10]) is essential for mediating binding of SN25 to Rph3A FL. However, previous structural and biochemical analysis using the $C_2B$ fragment of Rph3A suggested the middle portion of SN1 (E38/D41/R45) as the primary binding sites for Rph3A (*Ferrer-Orta et al., 2017*). To figure this out, we made an SN25 (1–82) mutant which carries E38A/D41A/R45A (EDR) and examined its interaction with Rph3A FL. Similar to SN25 (1–82), the EDR mutant retained strong binding to Rph3A FL (*Figure 2—figure supplement 1*), suggesting that the N-peptide rather than the SN1 mediates Rph3A FL interaction. Taken together, these data indicate that the binding mode between Rph3A FL and SN25 is distinct from the mode between Rph3A-$C_2B$ and SN25, in line with the results shown in *Figure 1*. In addition, sequence alignment showed that the negatively charged N-peptide sequence of SN25 is conserved among different species (*Figure 2—figure supplement 2A*). However, the N-peptide sequence of SN25 is not conserved in other isoforms such as SNAP-23, SNAP-47, SNAP-29, and Qb SNAREs (e.g., Vti1A, Vti1B, GOSR1, GOSR2, and Sec20) (*Figure 2—figure supplement 2B,C*). To identify the key residues in the sequence that mediates Rph3A interaction, we introduced the E3A, D4A, and D6A single mutation to SN25 (1–82), and found that the D4A mutation significantly impaired binding of SN25 (1–82) to Rph3A FL (*Figure 2—figure supplement 3*). Moreover, we verified this binding modality in the presence of liposomes with physiological lipid composition, we performed lipid sedimentation assay and found that compared to SN25 (1–206), deletion of the N-peptide of SN25 (11–206) remarkably impaired binding to Rph3A FL in the presence of negative membranes (*Figure 2—figure supplement 4*). Altogether, these results show that SN25 N-peptide mediates Rph3A FL interaction.

## The N-peptide of SN25 is essential for vesicle docking and fusion in PC12 cells

It was previously reported that Rph3A regulates exocytosis of DCV in neuroendocrine cells (*Tsuboi and Fukuda, 2005*). We aimed to explore whether the N-peptide of SN25 is required for DCV docking and fusion by using knockdown-rescue approach in cultured PC12 cells. Endogenous SN25 expression was strongly suppressed by virally delivered shRNAs (see Materials and methods) as previously described (*Cahill et al., 2006*; *Zhou et al., 2019*; *Wang et al., 2020*; *Figure 3A*). In parallel, neuropeptide Y (NPY)-td-mOrange2 was applied to visualize DCVs. First, we monitored the dynamics of NPY-td-mOrange2-labeled DCVs near the plasma membrane by TIRF microscopy and counted the number of plasma membrane-associated DCVs in resting PC12 cells. As expected, SN25 knockdown cells showed significantly reduced number of plasma membrane-docked DCVs, in comparison with control cells (*Figure 3B and C*). Expression of SN25 (full length, 1–206) restored the number of docked DCVs, whereas the expression of the SN25 mutant (11–206) that lacks the N-peptide exhibited strongly impaired ability to restore the docked DCVs in PC12 cells (*Figure 3B and C*), revealing that the N-peptide of SN25 is required for DCV docking. Next, we analyzed the NPY-td-mOrange2 release events in PC12 cells under high KCl stimulation. The total number of release events in SN25 knockdown cells was significantly fewer than that in control cells (*Figure 3D and E*). Consistent with the docking phenotypes, expression of SN25 rescued the DCV release events, but expression of the SN25 mutant (11–206) failed to do so (*Figure 3D and E*). Taken together, these results indicate that SN25 N-peptide-mediated Rph3A interaction is required for DCV docking and fusion of exocytosis in PC12 cells.

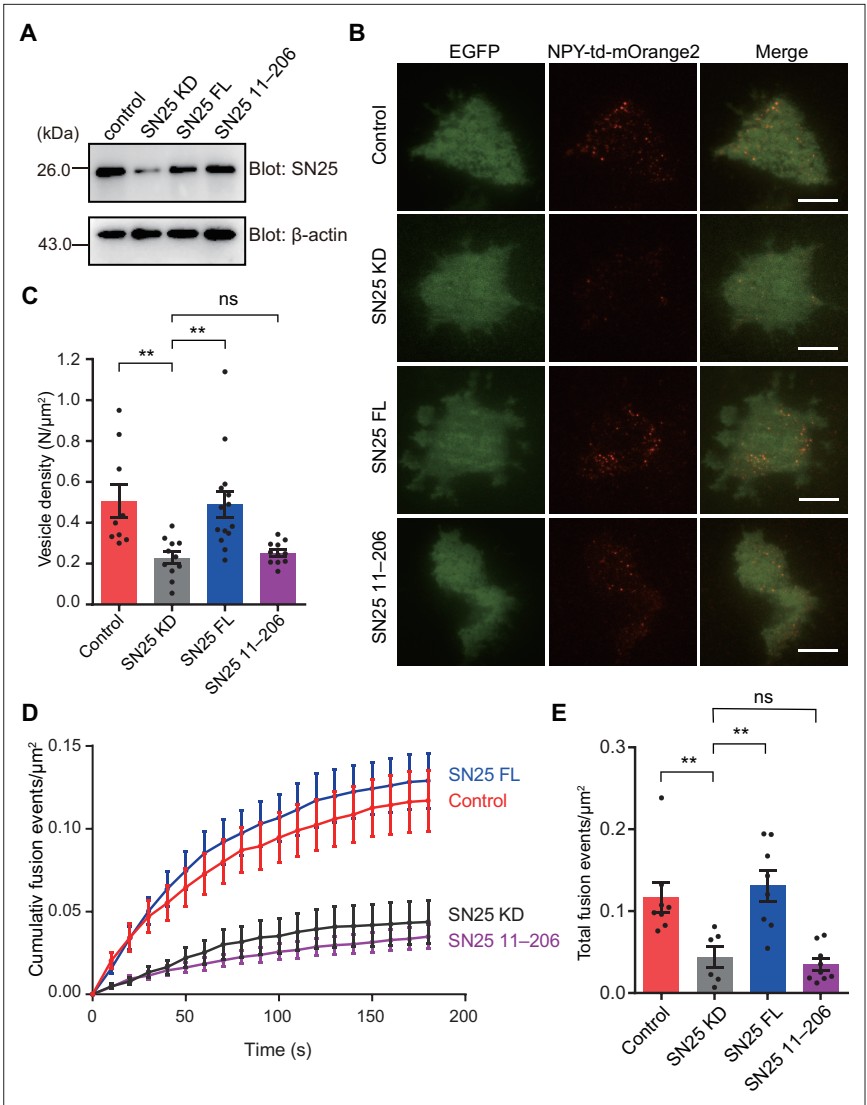

**Figure 3.** Importance of the N-peptide SN25 for DCV exocytosis in PC12 cells. (**A**) Immunoblotting assay to determine the SN25 expression level in control, SN25 KD, SN25 FL, or SN25 (11–206) rescued PC12 cells. Expression of SN25 and β-actin in PC12 cells was analyzed by 15% SDS-PAGE followed by immunoblotting with anti-SN25 antibody, and anti-β-actin antibody, respectively. (**B**) Representative TIRF images of control, SN25 KD, SN25 FL, or SN25 (11–206) rescued PC12 cells expressing indicator EGFP, and DCV content (NPY-td-mOrange2) are shown. Scale bars, 10 μm. (**C**) The density of docked vesicles was determined by counting the NPY-td-mOrange2 labeled vesicles in each image (n≥9 cells in each). (**D**) NPY-td-mOrange2 release events detected by TIRF microscopy during sustaining high K[+] stimulation. The curves indicate the cumulative number of fusion events per μm$^2$ in each cell. (**E**) Quantification of the results at 180 s in the experiments of (**D**). Data are presented as mean ± SEM; (n≥6 cells in each). Statistical significance and p values were determined by one-way analysis of variance (ANOVA). **p<0.01; ns, not significant.

The online version of this article includes the following source data for figure 3:

**Source data 1.** Uncropped Western blot shown in *Figure 3A*.

**Source data 2.** Excel file with data used to make *Figure 3C–E*.

## The bottom α-helix of the C$_2$B domain in Rph3A FL mediates interaction with SN25

The bottom α-helix (interface I) and the β3-β4 polybasic region (interface II) in the C$_2$B domain of Rph3A (*Figure 4A*), both having positively charged lysine residues, were implicated to mediate SN25 interaction (*Tsuboi et al., 2007*; *Ferrer-Orta et al., 2017*). Here, in the context of Rph3A FL, we

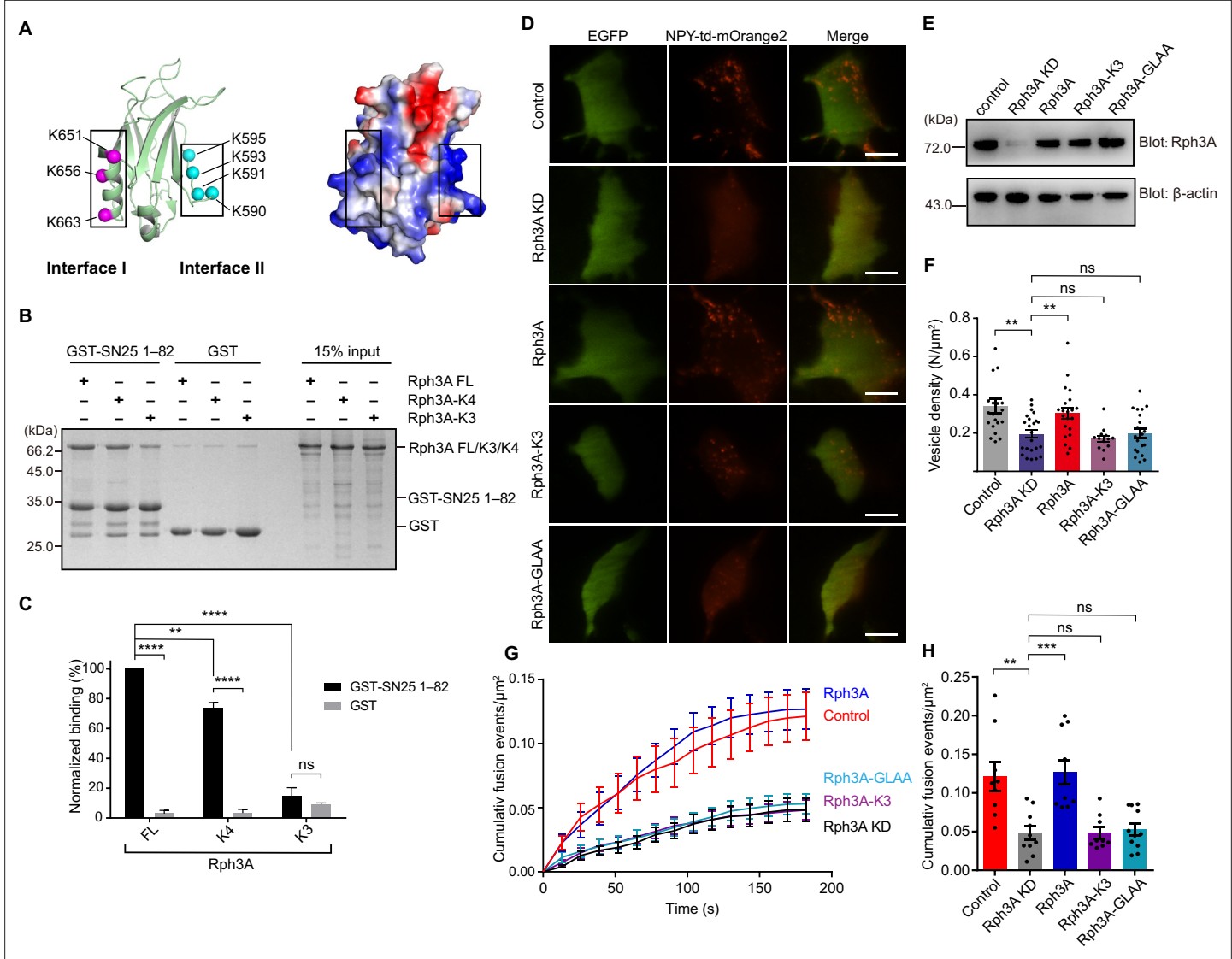

**Figure 4.** Importance of the C$_2$B bottom α-helix of Rph3A FL for SN25 binding and DCV exocytosis in PC12 cells. (**A**) Structural diagrams (left) and electrostatic surface potential (right) of Rph3A C$_2$B (PDB entry: 5LOW). Interface I residues K651/K656/K663 on the bottom α-helix and interface II residues K590/K591/K593/K595 on the side are shown as magenta and cyan spheres, respectively. Black boxes display the basic patches that include the residues shown on the left. (**B, C**) Binding of Rph3A FL or mutations to the SN25 (1–82) measured by GST pull-down assay (**B**) and quantification of the Rph3A binding (**C**). K3, Rph3A FL protein bearing the K651A/K656A/K663A mutation in interface I; K4, Rph3A FL protein bearing the K590Q/ K591Q/K593Q/K595Q mutation in interface II. Data are processed by ImageJ (NIH) and presented as the mean ± SEM (n=3), technical replicates. (**D**) Representative TIRF images of control, Rph3A KD, Rph3A WT, K3 mutant, or GLAA mutant rescued PC12 cells expressing indicator EGFP, and DCV content (NPY-td-mOrange2) are shown. Scale bars, 10 μm. (**E**) Immunoblotting assay to determine the Rph3A WT, Rph3A-K3, or Rph3A GLAA mutant expression level in PC12 cells. Expression of Rph3A WT or mutants and β-actin in PC12 cells was analyzed by 12% SDS-PAGE followed by immunoblotting with anti Rph3A antibody, and anti-β-actin antibody, respectively. The positions of the molecular mass markers are shown on the left. (**F**) The density of docked vesicles was determined by counting the NPY-td-mOrange2 labeled vesicles in each image (n≥13 cells in each). (**G**) NPY-td-mOrange2 release events detected by TIRF microscopy during sustaining high K$^+$ stimulation. The curves indicate the cumulative number of fusion events per μm$^2$ in each cell. (**H**) Quantification of the results at 180 s in the experiments of (**G**). Data are presented as mean ± SEM; (n≥9 cells in each). Statistical significance and p values were determined by one-way (in (**F**) and (**H**)) or two-way (in (**C**)) analysis of variance (ANOVA). **p<0.01; ***p<0.001; ****p<0.0001; ns, not significant.

The online version of this article includes the following source data and figure supplement(s) for figure 4:

**Source data 1.** Uncropped SDS-gel shown in *Figure 4B*.

**Source data 2.** Excel file with data used to make *Figure 4C and F–H*.

**Source data 3.** Uncropped Western blot shown in *Figure 4E*.

*Figure 4 continued on next page*

*Figure 4 continued*

**Figure supplement 1.** Effect of Rph3A interface I mutant on SN25 (1–140) binding.

**Figure supplement 1—source data 1.** Uncropped SDS-gel shown in *Figure 4—figure supplement 1A*.

**Figure supplement 1—source data 2.** Excel file with data used to make *Figure 4—figure supplement 1B*.

sought to figure out which interface is physiologically relevant and represents the actual binding site for SN25. We first examined the interactions between the two interfaces and SN25 (1–82) in vitro. Accordingly, we designed two mutations in Rph3A FL, that is, the K651A/K656A/K663A mutation (termed K3) in interface I and the K590Q/K591Q/K593Q/K595Q mutation (termed K4) in interface II (*Figure 4A*). Using GST pull-down assay, we found that K4 showed slightly reduced binding to SN25 (1–82) compared to Rph3A FL, whereas K3 displayed remarkably impaired interaction with SN25 (1–82) (*Figure 4B and C*). Similar results were found when the mutations were introduced to SN25 (1–140) (*Figure 4—figure supplement 1*). These data indicate that the bottom α-helix rather than the β3-β4 polybasic region in the $C_2B$ domain of Rph3A mediates SN25 interaction.

We next determined the functional importance of the bottom α-helix in DCV exocytosis in PC12 cells by using knockdown-rescue approach. Endogenous Rph3A expression was suppressed efficiently by virally delivered shRNAs (see Materials and methods) (*Figure 4E*). We then monitored the dynamics of NPY-td-mOrange2 labeled vesicles near the plasma membrane by TIRF microscopy as described in *Figure 3*. Rph3A KD significantly reduced the number of plasma membrane-docked DCVs in PC12 cells, in comparison with control cells (*Figure 4D and F*). Expression of Rph3A restored DCVs docking (*Figure 4D and F*). Consistent with in vitro binding data, expression of the Rph3A-K3 mutant could not restore the number of docked DCVs (*Figure 4D and F*). The Rph3A-GLAA mutant which disrupts the intramolecular interaction between Rph3A N- and C-terminal also failed to restore DCVs docking in PC12 cells (*Figure 4D and F*). These results suggest that the interaction between SN25 and Rph3A with proper configuration is important for DCVs docking. Similarly, the total number of release events in Rph3A knockdown cells was significantly fewer than that in control cells under high KCl stimulation (*Figure 4G and H*). In line with the docking phenotypes, expression of Rph3A but bot Rph3A-K3 mutant or Rph3A-GLAA mutant rescued the number of DCV release events (*Figure 4G and H*). Altogether, these results demonstrate that the Rph3A–SN25 interaction mediated by the $C_2B$ bottom α-helix is essential for DCV docking and fusion of exocytosis in PC12 cells.

## Rph3A FL accelerates SNARE complex assembly via SN25 interaction

We next investigated how the Rph3A–SN25 interaction regulates DCV exocytosis. We explored the influence of the Rph3A–SN25 interaction on SNARE complex assembly by using fluorescence resonance energy transfer (FRET) assay as previously described (*Yang et al., 2015*). Fluorescence-labeled Syb2 (29–93, S61C-BODIPY FL [BDPY], donor) and SN25 (1–206 or 11–206, R59C-Tetramethylrhodamine-5-maleimide [TMR], acceptor) were added with Syx1 (2–253), and SNARE complex assembly was detected by monitoring the decreased fluorescent signal of Syb2-BDPY due to FRET between fluorescence-labeled Syb2 and SN25 (*Figure 5A*). Intriguingly, we found that Rph3A FL remarkably accelerated SNARE complex assembly (*Figure 5B and C*). In contrast, the Rph3A-K4 mutant that maintains SN25 interaction accelerated the assembly as effectively as Rph3A FL, while the Rph3A-K3 and GLAA mutants that impair SN25 binding failed (*Figure 5B and C*). Hence, the assembly results suggest that the $C_2B$ bottom α-helix of Rph3A promotes SNARE complex assembly via SN25 binding.

Since our data revealed that the N-peptide of SN25 mediates Rph3A FL interaction, we then explored the importance of the N-peptide of SN25 for the acceleration of SNARE complex assembly by Rph3A FL. Indeed, in contrast to SN25 FL, SN25 (11–206) that lacks the N-peptide strongly impaired Rph3A activity in accelerating the assembly (*Figure 5D and E*), demonstrating that Rph3A FL promotes SNARE complex assembly via binding of the $C_2B$ bottom α-helix to the N-peptide of SN25.

To explore the role of Rph3A–SN25 interaction in trans-SNARE complex assembly, we used SNAREs reconstituted on liposomes. Liposomes bearing Syx1 (1–288) and liposomes containing Syb2 (29–116, S61C-BODIPY FL [BDPY], fluorescent donor) were mixed in the presence of SN25 Δ9 (1–197, deletion of the nine residues in the C-terminal end of SN25, R59C-Tetramethylrhodamine-5-maleimide [TMR], fluorescent acceptor); SN25 Δ9 is able to assemble into the trans-SNARE complex

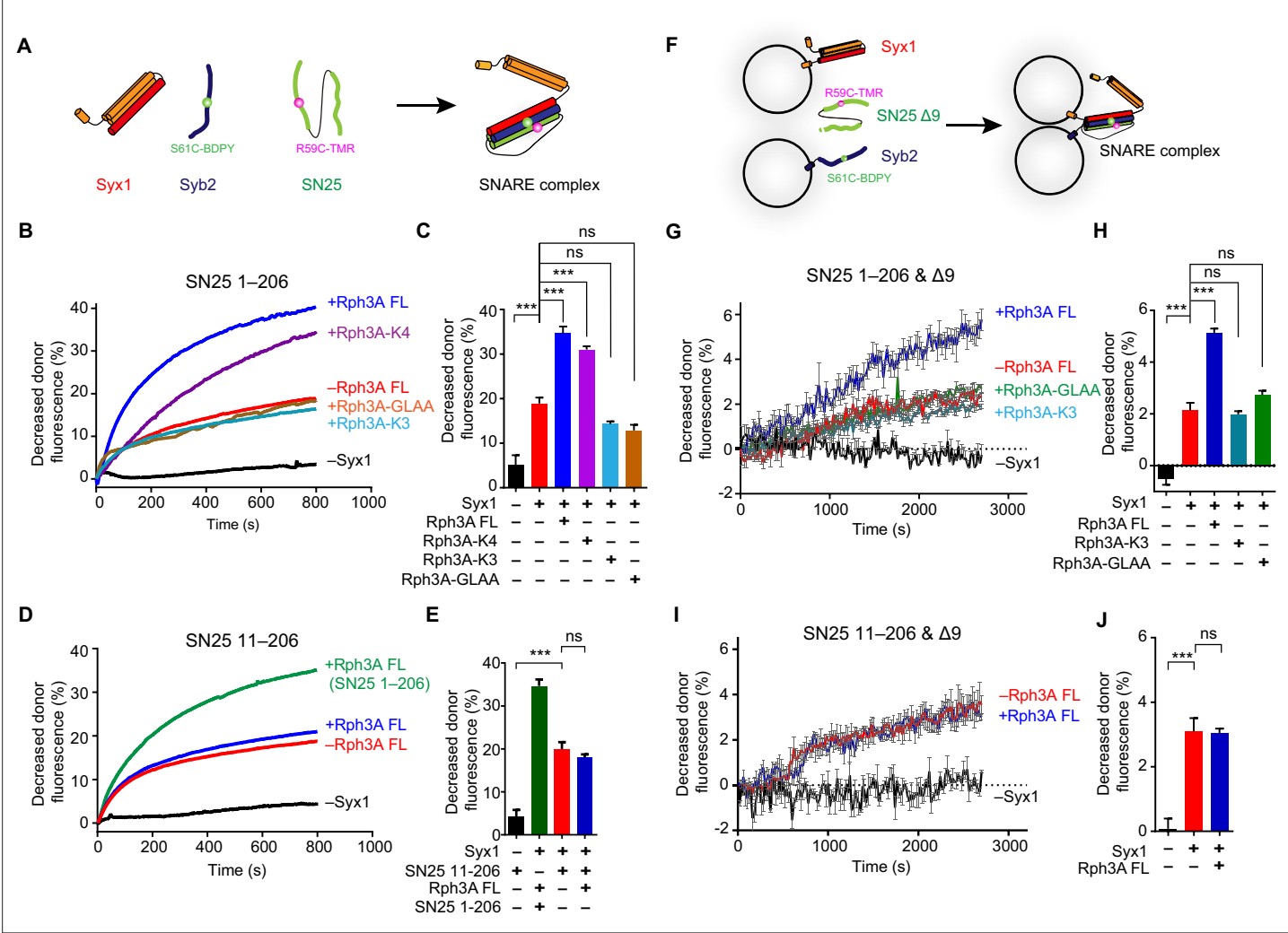

**Figure 5.** The function of Rph3A in SNARE complex assembly (**A**) Schematic diagram of SNARE complex assembly in the presence of Syx1 (2–253), Syb2 (29–93, S61C), and SN25 FL (1–206, R59C) or SN25 fragment (11–206, R59C). The FRET signal between Syb2 S61C-BDPY (donor) and SN25 R59C-TMR (acceptor) was monitored. (**B**) Effects of the K3 and/or K4 mutations on Rph3A-promoted SNARE complex assembly detected by FRET assay. (**C**) Quantification of the results at 600 s in the experiments of (**B**). (**D**) Effects on SN25 (11–206) on Rph3A-promoted SNARE complex assembly detected by FRET assay. Curve Rph3A (SN25 FL) (green color) represents the SN25 FL mediated SNARE complex assembly in the presence of Rph3A FL. (**E**) Quantification of the results at 600 s in the experiments of (**D**). (**F**) Schematic diagram of the trans-SNARE complex formation between Syx1 liposomes and Syb2 liposomes in the presence of SN25 and Rph3A FL or K3 and GLAA mutants. BDPY and TMR, which act as fluorescent donor and acceptor as above, were separately labeled to Syb2 and SN25 Δ9 or SN25 (11–206 & Δ9) truncation, respectively. (**G**) FRET assay for monitoring trans-SNARE complex assembly on membranes in the presence of the SN25 Δ9 and Rph3A FL or K3 and GLAA mutants. (**H**) Quantification of the results at 2300s in the experiments of (**G**). (**I**) FRET assay for monitoring trans-SNARE complex assembly on membranes in the presence of the SN25 (11–206 & Δ9) and Rph3A FL. (**J**) Quantification of the results at 2500s in the experiments of (**I**). Data are presented as mean ± SEM (n=3), technical replicates. Statistical significance and p values were determined by one-way analysis of variance (ANOVA). \*\*\*p<0.001; ns, not significant. FRET, fluorescence resonance energy transfer.

The online version of this article includes the following source data and figure supplement(s) for figure 5:

**Source data 1.** Excel file with data used to make *Figure 5C, E and G–J*.

**Figure supplement 1.** The function of Rph3A in SNARE-mediated membrane fusion.

**Figure supplement 1—source data 1.** Excel file with data used to make *Figure 5—figure supplement 1C,E,G*.

without inducing membrane fusion (*Lu, 2015*). Trans-SNARE complex assembly was detected as described in *Figure 5F*. As observed in *Figure 5G and H*, Rph3A FL accelerated trans-SNARE complex assembly, but the K3 and GLAA mutants failed to accelerate the assembly, indicating that the $C_2B$ bottom α-helix of Rph3A promotes trans-SNARE complex assembly via SN25 binding. In

addition, unlike SN25 Δ9, SN25 (11–206 & Δ9) lacking the N-peptide failed to support Rph3A activity in accelerating trans-SNARE complex assembly (*Figure 5I and J*), reinforcing the importance of the interaction between Rph3A C$_2$B bottom α-helix and SN25 N-peptide in trans-SNARE complex assembly.

Moreover, we investigated the function of Rph3A–SN25 interaction in membrane fusion using lipid mixing assay as previously described (*Wang and Ma, 2022*; *Figure 5—figure supplement 1A*). Indeed, we found that Rph3A FL promoted SNARE-mediated lipid mixing, but the Rph3A-K3 mutant failed (*Figure 5—figure supplement 1B,C*). In addition, SN25 (11–206) lacking the N-peptide failed to support Rph3A activity in lipid mixing, in comparison with SN25 FL (*Figure 5—figure supplement 1D,E*). Similarly, due to the N-peptide residues not conserved with SN25, SN23 also failed to support Rph3A activity in lipid mixing (*Figure 5—figure supplement 1F,G*). These results suggest that Rph3A promotes membrane fusion dependent on its interaction with the N-peptide of SN25.

## Rph3A FL induces conformational change in the SN1 domain of SN25

Next, we explored the mechanism how the Rph3A–SN25 interaction promotes SNARE complex assembly. As the N-peptide of SN25 mediates Rph3A FL interaction, we suspect whether this interaction would induce potential conformational changes of the SNARE motif (SN1) adjacent to the N-peptide. To this aim, we used a bimane-tryptophan quenching assay to analyze conformational changes of SN1 in SN25, as this assay has been widely used to study the structure and movement of proteins and has shown strong sensitivity in short-distance electron transfer measurements (10 Å) (*Mansoor et al., 2002*; *Islas and Zagotta, 2006*; *Taraska and Zagotta, 2010*). In this case, a single tryptophan mutation (E55W) located at the middle portion of the SN1 was introduced in SN25 (full length, 1–206), and a single cysteine mutation (R59C) adjacent to E55W was created for bimane labeling. Note that the E55W/R59C double mutation did not influence the activity of SN25 in SNARE complex assembly (*Figure 6—figure supplement 1*). If a conformational change occurred in SN1, for instance, a transition from random coils to α-helix, this change would shorten the distance between E55W and R59C-bimane thus produces enhanced bimane-tryptophan quenching (*Figure 6A*). In the original state of SN25 (unstructured state), SN25 E55W/R59C exhibited remarkably reduced bimane fluorescence compared to SN25 R59C (*Figure 6—figure supplement 2A*), indicating efficient electron transfer in the distance between E55W and R59C. As a positive control, addition of Syx1 and Syb2 to SN25 E55W/R59C led to further reduced bimane fluorescence (*Figure 6B and C*), consistent with the conformational change of SN1 (from random coils to α-helix) driven by the formation of the SNARE complex (*Sutton et al., 1998*). These data confirmed that conformational change of SN1 can be detected by using the bimane-tryptophan quenching assay. Intriguingly, when Rph3A FL was added to SN25 E55W/R59C in the absence of Syx1 and Syb2, we observed further reduced bimane fluorescence (*Figure 6B and C*), similar to that observed with the addition of Syx1 and Syb2. Hence, these results indicate that Rph3A is able to induce a conformational change of SN1 independent of Syx1 and Syb2.

Then, we examined whether Rph3A FL-induced conformational change of SN1 is mediated by the interaction between the C$_2$B bottom α-helix of Rph3A and the N-peptide of SN25. In this case, the Rph3A-K3 mutant and SN25 (11–206) were accordingly used in the bimane-tryptophan quenching assay. In contrast to Rph3A, addition of the Rph3A-K3 mutant to SN25 E55W/R59C failed to further reduce bimane fluorescence (*Figure 6B and C*), indicating the requirement of the C$_2$B bottom α-helix of Rph3A. To verify the importance of the N-peptide of SN25, we created SN25 (11–206) R59C and SN25 (11–206) E55W/R59C (*Figure 6D*). As expected, SN25 (11–206) E55W/R59C exhibited reduced bimane fluorescence compared to SN25 (11–206) R59C (*Figure 6—figure supplement 2B*). Likewise, further reduced bimane fluorescence was observed for SN25 (11–206) E55W/R59C upon the addition of Syx1 and Syb2. However, in sharp contrast to SN25 E55W/R59C, SN25 (11–206) E55W/R59C displayed no further reduced bimane fluorescence upon the addition of Rph3A FL in the absence of Syx1 and Syb2 (*Figure 6E and F*), indicating the importance of the N-peptide of SN25. Taken together, our data suggest that Rph3A FL promotes SNARE complex assembly via inducing the conformational change of SN1 from random coil to α-helix, and the promotion requires the interaction of the C$_2$B bottom α-helix of Rph3A with the N-peptide of SN25.

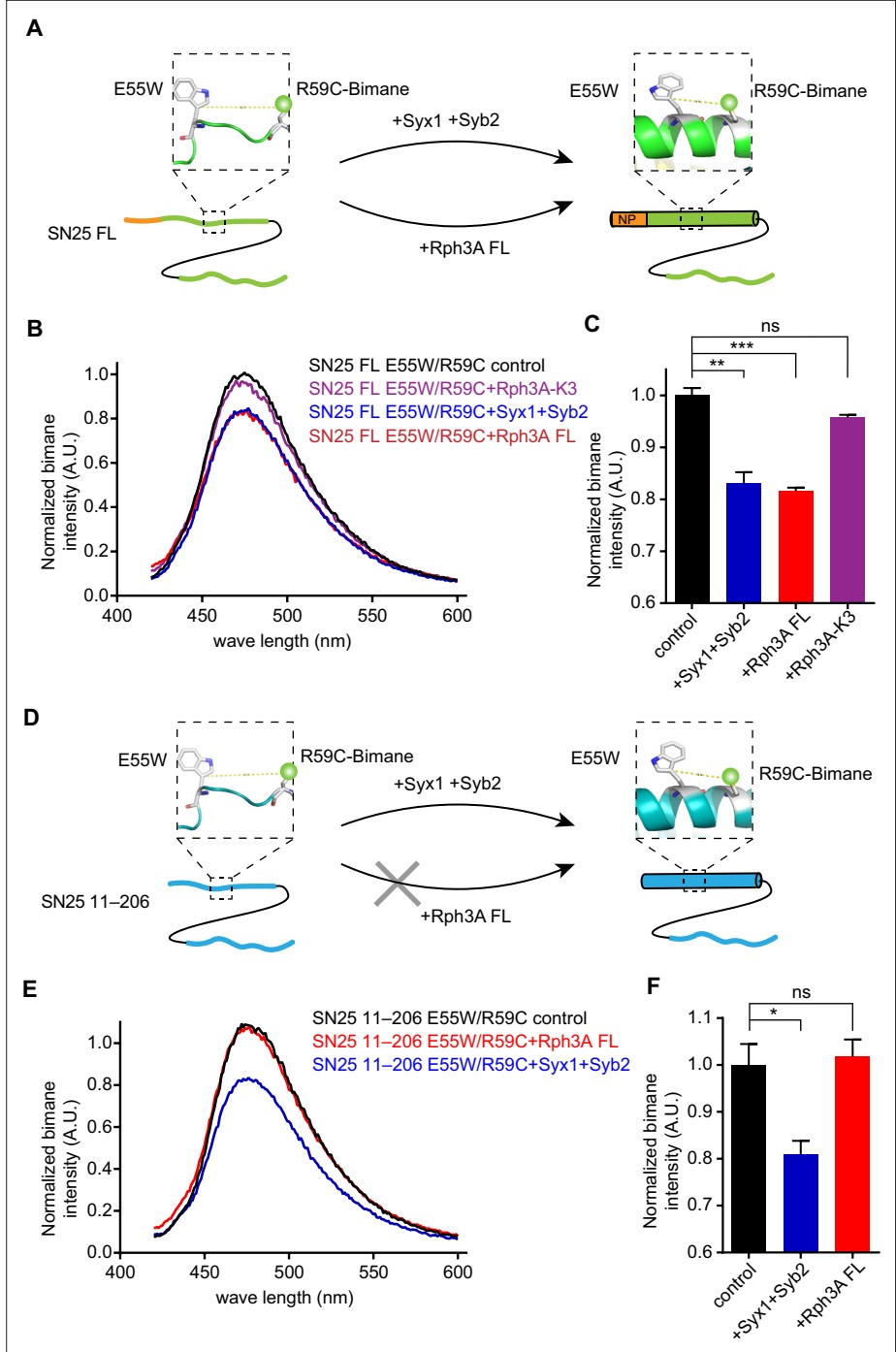

**Figure 6.** Conformation change of SN25 induced by Rph3A. (**A, D**) Illustration of the conformation change from random coils to α-helix in the SN1 of SN25 induced by Syx1 (2–253) and Syb2 (29–93), or by Rph3A, monitored by bimane-tryptophan quenching assay. Tryptophan was introduced at residue E55 and bimane fluorescence was labeled on residue R59C in SN25 FL (**A**) or SN25 11–206 (**D**). (**B, C**) Quenching of bimane fluorescence on the SN25 E55W/R59C with the addition of Syx1 and Syb2, and the addition of Rph3A FL or Rph3A-K3 mutant (**B**) and quantification of the results observed at 474 nm (**C**). (**E, F**) Quenching of bimane fluorescence on the SN25 (11–206) E55W/R59C with the addition of Syx1 and Syb2, or the addition of Rph3A FL (**E**) and quantification of the results observed at 474 nm (**F**). Data are presented as mean ± SEM (n=3), technical replicates. The significance were examined by one-way analysis of variance (ANOVA). *p<0.05; **p<0.01; ***p<0.001; ns, not significant.

The online version of this article includes the following source data and figure supplement(s) for figure 6:

*Figure 6 continued on next page*

*Figure 6 continued*

**Source data 1.** Excel file with data used to make *Figure 6C and F*.

**Figure supplement 1.** Assembly of the SNARE complex measured by fluorescence anisotropy.

**Figure supplement 2.** The intrinsic bimane fluorescence quenching by E55W mutant on SN25.

**Figure supplement 2—source data 1.** Excel file with data used to make *Figure 6—figure supplement 2*.

## Discussion

Rph3A was initially identified as a specific Rab3A/Rab27A-binding protein on secretory granules that are involved in the control of regulated vesicle secretion, including neurotransmitter release and hormone secretion (*Li et al., 1994*; *Fukuda et al., 2004*). Rph3A was later found to bind phospholipids via its $C_2$ domains and interact with SN25 and phosphatidylinositol-4,5-bisphosphate (PI(4,5) P2) via its $C_2B$ domain (*Yamaguchi et al., 1993*; *Chung et al., 1998*; *Ferrer-Orta et al., 2017*). In particular, the $C_2B$ domain of Rph3A was generally regarded as the pivotal unit to bind SN25, and this interaction was implicated to regulate docking and fusion of DCVs in neuroendocrine cells and to control repriming of synaptic vesicles in neurons (*Tsuboi and Fukuda, 2005*; *Deák et al., 2006*). Despite these advances, the physiologically relevant binds sites between Rph3A and SN25 remain elusive, and how the Rph3A–SN25 interaction regulates exocytosis has not been fully elucidated. In the present study, we found a new binding mode between Rph3A and SN25, and revealed its importance in regulating DCV exocytosis in PC12 cells. Our data highlight that this binding mode is essential for Rph3A activity in promoting SNARE complex assembly via a mechanism involving the conformational change of SN25.

The Rph3A $C_2B$ domain was previously identified to bind SN25 (*Tsuboi and Fukuda, 2005*; *Deák et al., 2006*), but the physiologically relevant binding sites remain elusive. Our present data for the first time identified a Rph3A–SN25 interaction that relies on the intramolecular interplay between the RBD and $C_2AB$ of Rph3A (*Figure 1*), suggesting that both domains act synergistically to bind SN25. In addition, this interaction is compatible with Rab3A interaction, indicative of the formation of a Rab3A–Rph3A–SN25 ternary complex essential for bridging vesicles and the plasma membrane (*Figure 7*). Besides, considering that the PRL of Rph3A connecting the RBD and $C_2AB$ domains contain multiple sites which can be phosphorylated by PKA, PKC, and/or CaMKII (*Kato et al., 1994*; *Numata et al., 1994*; *Fykse et al., 1995*; *Foletti et al., 2001*), it is plausible that phosphorylation of the PRL is able to alter the Rph3A–SN25 interaction via influencing the intramolecular interplay of the RBD with $C_2AB$, which modulates Rph3A activity in vivo.

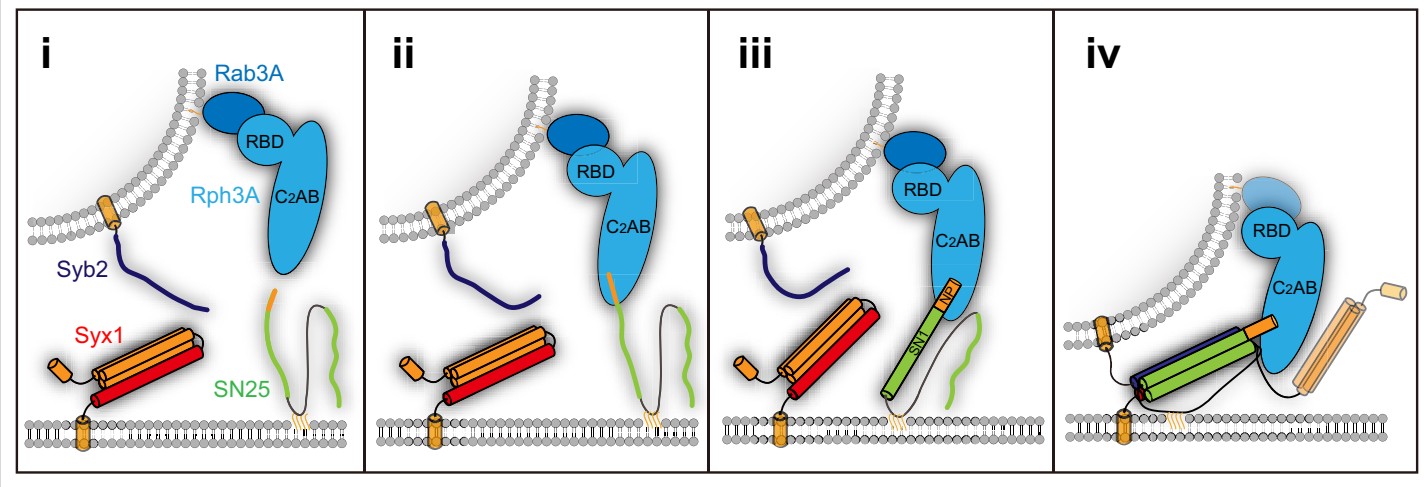

**Figure 7.** Model of Rph3A function in prefusion steps of exocytosis. Stage i, Rph3A is associated with trafficking vesicles via binding to GTP-Rab3A. Stage ii, Rph3A promotes vesicle docking via binding to plasma membrane-bound SN25 and vesicle-bound Rab3A together. Stage iii, upon binding to the N-peptide of SN25, Rph3A induces a conformational change of SN1 from random coils to α-helix. Stage iv, Rph3A accelerates SNARE complex assembly via SN25 interaction, which promote vesicle priming. As indicated, multiple SNARE regulatory proteins, for example, Munc18-1, Munc13-1, Doc2, and Syt1 are also involved in stages i–iv.

In addition to the SNARE motifs of SN25 involved in the formation of the SNARE four-helix bundle, other sequences in SN25 have gained attention in recent years. For instance, the linker that connects SN1 and SN2 has been suggested to act as a flexible molecular spacer to ensure efficient S-acylation and support fusion intermediates by local lipid interactions (*Salaun et al., 2020*; *Shaaban et al., 2019*), or recruited by Munc13-1 to chaperone SNARE complex assembly (*Kalyana Sundaram et al., 2021*). In the present study, we identified that the N-peptide sequence (1–10) rather than the SNARE motifs of SN25 mediates interaction with Rph3A FL (*Figure 2*), and this binding specificity might be conferred by synergistic action of the RBD with the C$_2$AB of Rph3A as discussed above.

Actually, according to the recently solved Rph3A-C$_2$B/SN25 (Δ1–6) structure (*Ferrer-Orta et al., 2017*), three distinct regions in SN25 were shown to participate in binding to the bottom α-helix of Rph3A C$_2$B domain, one involved R8/E10/Q15 in the N-terminal peptide of SN25, the other two included E38/D41/R45 and D51/E55/R59 in the middle of SN1, respectively. However, these binding information need to be interpreted with caution, because in previous work (i) the isolated C$_2$B domain, but not Rph3A FL was employed; (ii) the N-peptide residues of SN25 were partially deleted; and (iii) physiological relevance of these interactions had not been tested yet. Our observations that Rph3A FL bound efficiently to SN25 E38A/D41A/R45A mutant and D51A/E55A/R59A mutant but not to SN25 lacking the N-peptide pinpoint the binding specificity between Rph3A FL and the SN25 N-peptide (*Figure 2E* and *Figure 2—figure supplement 1*). Consistently, deletion of the N-peptide of SN25 or disruption of the C$_2$B bottom α-helix of Rph3A significantly impaired docking and fusion of DCVs with the plasma membrane in PC12 cells (*Figures 3 and 4*), revealing the physiological relevance of such binding mode.

Doc2 and Synaptotagmin-1(Syt1), which contain tandem C$_2$ domains homologous to that of Rph3A, are major Ca$^{2+}$ sensors in neurons responsible for spontaneous, asynchronous, and synchronous release of neurotransmitters (*Kaeser and Regehr, 2014*). Doc2 and Syt1 were both implicated to bind SN25 at the middle of SN1 (D51/E52/E55) (*Groffen et al., 2010*; *Brewer et al., 2015*; *Zhou et al., 2015*). In this circumstance, the preference for binding of Rph3A to the N-peptide of SN25 identified here could make D51/E52/E55 more available for Doc2 and/or Syt1 interaction. In addition, interestingly, our results found for the first time that Rph3A has an ability to promote SNARE complex assembly, which requires binding of the C$_2$B bottom α-helix of Rph3A to the N-peptide of SN25 (*Figure 5*). As indicated by bimane-tryptophan quenching assay (*Figure 6*), our results found that Rph3A induces a conformation change in the SNARE motif of SN25, which increases the content of SN1' α-helical configuration to facilitate its assembly with the SNARE motifs of Syx1 and Syb2. Taken together, we suggest that Rph3A has at least two important roles in prefusion events of exocytosis: one is to promote vesicle docking dependent on its associations with Rab3A and SN25 between the two opposite membranes; the other is to promote vesicle priming via accelerating SNARE complex assembly (*Figure 7*). These mechanisms may partly underlie the functional importance of Rph3A in DCV exocytosis in PC12 cells.

Previous studies have revealed that the N-peptide of Syx1 bound to Munc18-1 plays a regulatory role in exocytosis and that the N-terminal proline-rich sequence of Syb2 bound to intersectin-1 is necessary for the clearance of SNARE complexes required for vesicle recycling (*Rathore et al., 2010*; *Jäpel et al., 2020*). Combined with these results, the functional importance of the N-peptide of SN25 bound to Rph3A identified here suggests a common feature that the N-terminal regions of the three SNAREs are required for modulating vesicle exocytosis via interactions with multiple regulatory proteins.

In isolated form, the SNARE motifs of Syx1, SN25, and Syb2 all assume unstructured conformation (*Fasshauer et al., 1997*; *Hazzard et al., 1999*). However, as observed in the crystal structures of Munc18-1 bound to Syb2 or Syx1 and of Vps33 bound to Nyv1 or Vam3, the SNARE motifs of Syb2 or Syx1, and Nyv1 or Vam3 display highly ordered α-helical structures (*Baker et al., 2015*; *Stepien et al., 2022*), resemblance to their conformations in the assembled SNARE complex, suggesting that the prestructured SNARE motif of the SNAREs might be essential for proper SNARE pairing and register. Highly consistent with this notion, our data showed that Rph3A bound to SN25 renders SN1 to assume a prestructured conformation (*Figure 6*). These evidence suggests that the prestructured state of the three SNAREs induced by multiple regulatory proteins might be a prerequisite for the efficiency of SNARE complex assembly in exocytosis.

In contrast to the strong phenotype of Rph3A loss as observed in neuroendocrine cells, Rph3A loss has almost no phenotype in mammalian neurons (*Schlüter et al., 1999*). It is of note that the synapse has evolved a number of specialized sites beneath the presynaptic membrane (active zones) opposing the postsynaptic density, which are composed of multiple active-zone components required for synaptic vesicle exocytosis. Such active-zone components may take the place of Rph3A in tethering/docking vesicles at the release sites and priming them for fusion. For example, Rab-binding protein RIMs and other vesicle associated proteins, such as Munc13 and CAPS, have been found to support vesicle tethering/docking via mediating the interaction between synaptic vesicles and the plasma membrane and to support vesicle priming via promoting SNARE complex assembly (*Dulubova et al., 2005*; *Imig et al., 2014*; *Quade et al., 2019*; *Wang et al., 2017*; *Wang et al., 2019*; *James et al., 2009*). However, neuroendocrine cells lack compartmentalized active zones on the plasma membrane, in which RIMs, Munc13, CAPS, and so forth, are dominantly distributed in the cytosol instead of the plasma membrane (*Fukuda, 2004*; *Kabachinski et al., 2014*; *Kabachinski et al., 2016*; *Houy et al., 2022*), rendering them unlikely to compensate the loss of Rph3A in DCV exocytosis. A second possibility is that other SN25 binding partners in synapses alternatively promote SN25 assembly into the SNARE complex. For example, Munc13-1 was recently found to bind the linker region of SN25 to accelerate SNARE complex assembly (*Kalyana Sundaram et al., 2021*). Hence, with multiple components enriched at the active zones, synapses may permit normal exocytosis in a manner independent of Rph3A.

# Materials and methods

## Key resources table

| Reagent type (species) or resource | Designation | Source or reference | Identifiers | Additional information |
|---|---|---|---|---|
| Strain, strain background (*Escherichia coli*) | BL21 (DE3) | Agilent | Cat#: 230245 | Strain for expressing recombinant proteins |
| Cell line (Rattus Norvegicus) | PC-12 | The cell bank of the typical culture deposit committee of Chinese academy of sciences | Cat#: TCR 8 | |
| Cell line (human) | Human embryo kidney (HEK) 293T | ATCC | Cat#: CRL-3216; RRID: CVCL_0063 | |
| Transfected construct (Rattus Norvegicus) | L309-shRph3A | This paper | | Lentiviral construct to transfect and express the shRNA for Rph3A knockdown |
| Transfected construct (human) | L309-shSN25 | This paper | | Lentiviral construct to transfect and express for SN25 knockdown |
| Transfected construct (Rattus Norvegicus) | L309-Rph3A | This paper | | Lentiviral construct to transfect and express the Rph3A |
| Transfected construct (Rattus Norvegicus) | L309-Rph3A-K3 | This paper | | Lentiviral construct to transfect and express the Rph3A-K3 |
| Transfected construct (Rattus Norvegicus) | L309-Rph3A-GLAA | This paper | | Lentiviral construct to transfect and express the Rph3A-GLAA |
| Transfected construct (human) | L309-SN25 | This paper | | Lentiviral construct to transfect and express the SN25 |
| Transfected construct (human) | L309-SN25 11–206 | This paper | | Lentiviral construct to transfect and express the SN25 11–206 |
| Antibody | Anti-β-actin (Mouse monoclonal) | Proteintech | Cat#: 66009-1-Ig; RRID: AB_11232599 | WB (1:10,000) |
| Antibody | Anti-His$_6$ (Mouse monoclonal) | Proteintech | Cat#: 66005-1-Ig; RRID: AB_11232599 | WB (1:10,000) |
| Antibody | Anti-SN25 (Rabbit polyclonal) | Proteintech | Cat#: 14903-1-AP; RRID: AB_2192051 | WB (1:5000) |
| Antibody | Anti-Rph3A (Rabbit polyclonal) | Proteintech | Cat#: 11396-1-AP; RRID: AB_2181145 | WB (1:1000) |
| Antibody | Anti-Mouse IgG(H+L) (Goat polyclonal) | Proteintech | Cat#: SA00001-1; RRID: AB_2722565 | WB (1:10,000) |

*Continued on next page*

*Continued*

| Reagent type (species) or resource | Designation | Source or reference | Identifiers | Additional information |
|---|---|---|---|---|
| Antibody | Anti-Rabbit IgG(H+L) (Goat polyclonal) | Proteintech | Cat#: SA00001-2; RRID: AB_2722564 | WB (1:10,000) |
| Recombinant DNA reagent | pEXP5-NT/TOPO-Rph3A FL | This paper | | 6xHis-tagged Rph3A FL (aa 1–681) for protein expression |
| Recombinant DNA reagent | pEXP5-NT/TOPO-Rph3A (Δ185–371) | This paper | | 6xHis-tagged Rph3A (Δ185–371) for protein expression |
| Recombinant DNA reagent | pEXP5-NT/TOPO-Rph3A (Δ185–371) | This paper | | 6xHis-tagged Rph3A (Δ185–371) for protein expression |
| Recombinant DNA reagent | pEXP5-NT/TOPO-Rph3A (Δ185–371) G102A/L104A | This paper | | 6xHis-tagged Rph3A (Δ185–371) G102A/ L104A for protein expression |
| Recombinant DNA reagent | pEXP5-NT/TOPO-Rph3A-K3 | This paper | | 6xHis-tagged Rph3A K3 mutant for protein expression |
| Recombinant DNA reagent | pEXP5-NT/TOPO-Rph3A-K4 | This paper | | 6xHis-tagged Rph3A K4 mutant for protein expression |
| Recombinant DNA reagent | pEXP5-NT/TOPO-Rph3A-GLAA | This paper | | 6xHis-tagged Rph3A GLAA mutant for protein expression |
| Recombinant DNA reagent | pET28a-Rph3A 1–281 | This paper | | 6xHis-tagged Rph3A 1–281 for protein expression |
| Recombinant DNA reagent | pET28a-Rph3A 182–681 | This paper | | 6xHis-tagged Rph3A 182–681 for protein expression |
| Recombinant DNA reagent | pET28a-Rph3A 282–681 | This paper | | 6xHis-tagged Rph3A 282–681 for protein expression |
| Recombinant DNA reagent | pET28a-Rph3A 372–681 | This paper | | 6xHis-tagged Rph3A 372–681 for protein expression |
| Recombinant DNA reagent | pET28a-SN25 | This paper | | 6xHis-tagged SN25 for protein expression |
| Recombinant DNA reagent | pET28a-SN25 11–206 | This paper | | 6xHis-tagged SN25 11–206 for protein expression |
| Recombinant DNA reagent | pGEX-6P-1-SN25 | This paper | | GST-tagged SN25 for protein expression |
| Recombinant DNA reagent | pGEX-6P-1-SN25 1–140 | This paper | | GST-tagged SN25 1–140 for protein expression |
| Recombinant DNA reagent | pGEX-6P-1-SN25 1–82 | This paper | | GST-tagged SN25 1–82 for protein expression |
| Recombinant DNA reagent | pGEX-6P-1-SN25 11–82 | This paper | | GST-tagged SN25 11–82 for protein expression |
| Recombinant DNA reagent | pGEX-6P-1-SN25 7–82 | This paper | | GST-tagged SN25 7–82 for protein expression |
| Software, algorithm | ImageJ | National Institutes of Health (NIH) | | |
| Software, algorithm | Prism 8.0.0 | GraphPad | | |
| Software, algorithm | Icy | BioImage Analysis unit Institut Pasteur | | |

## Cell lines

The HEK293T cell line used in this study was from the American Type Culture Collection and Thermo Fisher Scientific, authenticated by STR locus and tested negative for mycoplasma contamination. PC12 cell line was from the cell bank of the typical culture deposit committee of Chinese Academy of Sciences, authenticated by CSTR locus and tested negative for mycoplasma contamination.

## Plasmids and protein purification

The sequence encoding rat Syb2 (full length, residues 1–116), Syb2 (residues 29–93), Syb2 (residues 29–93, S61C), and Syb2 (residues 29–116, S61C) were cloned to pGEX-6p-1 (GE Healthcare, Piscataway, NJ). The sequence encoding mouse Rab3A (residues 22–217, Q81L) was cloned into pGEX-6p-1.

The sequence encoding rat Syx1 (residues 2–253) was cloned into pGEX-KG vector (GE Healthcare). The sequence encoding rat Syx1 (residues 1–288) was cloned into pET28a vector (Novagen, Australia). These proteins were expressed and purified as described (*Dulubova et al., 2005*; *Ma et al., 2013*; *Stepien et al., 2022*).

The sequence encoding human SN25 (full length, residues 1–206), SN25 fragments (residues 1–82, 1–140, 141–206, 83–206, 83–140, 7–82, and 11–82), and SN25 1–82 EDR mutant (E38A/D41A/R45A), DER mutant (D51A/E55A/R59A), E3A, D4A, and D6A mutant were cloned into pGEX-6P-1 vector (GE Healthcare). The sequence encoding rat Rph3A RBD (residues 40–170) and RBD G102A/L104A mutant were all cloned into pGEX-6P-1 vector. The sequence encoding human SN25 (residues 1–206, R59C), SN25 (residues 1–206, E55W/R59C), SN25 (residues 11–206, R59C), SN25 (residues 11–206, E55W/R59C), SN25 Δ9 (residues 1–197, R59C), SN25 11–206 and Δ9 (residues 11–197, R59C), and human SN23 (residues 1–211) were cloned into pET28a vector (Novagen, Australia). All the recombinant proteins were expressed in *E. coli* BL21 (DE3) strain cultured in LB media at 37°C to OD600 of 0.6–0.8 and were induced with 0.4 mM IPTG at 20°C for 16 hr. Cells were resuspended in a buffer containing 25 mM HEPES pH 7.4, 150 mM KCl, 10% glycerol (v/v). Cells were broken using AH-1500 Nano Homogenize Machine (ATS Engineering Inc) at 800 bars three times at 4°C. Cell lysates were centrifuged at 16,000 rpm for 30 min in a JA-25.50 rotor (Beckman Coulter) at 4°C. For the purification of GST fusion proteins, the supernatants were incubated with 2 ml glutathione-Sepharose beads (GE Healthcare) at 4°C for 3 hr. The bound proteins were eluted by a buffer containing 20 mM Tris pH 8.0, 150 mM NaCl, and 20 mM L-glutathione at 4°C for 3 hr. For the purification of His$_6$ fusion proteins, the supernatant was incubated with Ni$^{2+}$-NTA agarose (QIAGEN) at 4°C for 1 hr. The beads were washed with buffer 25 mM HEPES pH 7.4, 150 mM KCl, 10% glycerol (v/v) supplied with an additional 30 mM imidazole. The protein was eluted with a wash buffer described above but supplied with an additional 300 mM imidazole.

The sequence encoding rat full-length Rph3A (residues 1–681) was amplified from rat brain cDNA library and cloned into the pEXP5-NT/TOPO vector (Invitrogen). The sequence encoding rat Rph3A fragments (residues 1–281, 182–681, 282–681, and 372–681) were cloned into the pET28a vector. The sequence encoding rat Rph3A Δ185–371 (deletion residues 185–371), Rph3A Δ185–371 G102A/L104A, Rph3A Δ1–45 and Δ161–371 (deletion residues 1–45 and 185–371) were cloned into the pEXP5-NT/TOPO vector. The sequence encoding rat Rph3A-K3 mutant (K651A/K656A/K663A), Rph3A-K4 mutant (K590Q/K591Q/K593Q/K595Q), Rph3A-GLAA mutant (G102A/L104A) were also cloned into the pEXP5-NT/TOPO vector. For expression of full-length Rph3A and its truncation or mutants via pEXP5-NT/TOPO or pET28a vector, plasmids were transformed into *E. coli* BL21 (DE3). Cells were grown to OD600 of 0.6 in 37°C and IPTG was added to a final concentration of 0.2 mM. Cells were shaken overnight at 16°C for 18 hr, pelleted and resuspended in lysis buffer (25 mM HEPES pH 7.4, 1 M KCl). Cells were lysed by AH-1500 Nano Homogenize Machine (ATS Engineering Inc). Triton X-100 was added to a final concentration of 0.2% and PMSF to 1 mM. Lysates were incubated with 1 ml of Ni-NTA resin (QIAGEN) in 4°C for 1 hr. The resin was washed with 0.1 M citrate buffer (3 mM citric acid, 97 mM trisodium citrate, pH 6.3) containing 30 mM imidazole, and the bound Rph3A proteins were eluted with 300 mM imidazole in citrate buffer. All the eluted proteins were loaded into Superdex 200 pg or Superdex 75 pg size exclusion chromatography (GE Healthcare) to remove aggregates and potential contaminants.

The plasmid encoding NPY-td-mOrange2 was purchased from Addgene (plasmid no. 83497). For the SN25 knockdown in PC12 cells, the SN25 short hairpin RNA (shRNA) was designed as reported (*Cahill et al., 2006*) and cloned into pFHUUIG_shortU6 (L309) plasmid after the H1 promoter. The sequence encoding human SN25 FL (residues 1–206) or truncation fragment (residues 11–206) were cloned into pFHUUIG_shortU6 (L309) plasmid after the Ub promoter. For the Rph3A knockdown in PC12 cells, the Rph3A shRNA (Forward primer: 5′-TCGAG-CATTGGCAAATCGAATGATTATTCA AGAGATAAT-CATTCGATTTGCCAATGTTTTT-3′, reverse primer: 5′-CTAGAAAAACATTGGCAAAT C-GAATGATTATCTCTTGAATAATCATTCGATTTGCCAATGC-3′) was cloned into pFHUUIG_shortU6 (L309) plasmid after the H1 promoter as above. The sequence encoding rat Rph3A and Rph3A-K3 mutation or Rph3A-GLAA mutant were cloned into pFHUUIG_shortU6 (L309) plasmid after the Ub promoter.

## Analytical ultracentrifugation

To identify the aggregation state of Rph3A, sedimentation velocity (SV) experiments were performed using ProteomeLab XL-I analytical ultracentrifuge (Beckman Coulter, Palo Alto, CA). Rph3A proteins were prepared in 0.1 M Citrate buffer. The high purity proteins were collected and then concentrated to 1 OD (UV, 280 nm). Before the sample loading, a highly speed centrifuge (12,000 rpm) was done to remove the denatured protein aggregations. The reference cell was loaded with 400 µl 0.1 M Citrate buffer, and the sample cell was loaded with 380 µl Rph3A protein samples. Experiments were done with 40,000 rpm speed at 4°C overnight (An-60 Ti Rotor). Sedimentation profiles were recorded with UV (280 nm) absorbance and interference optics (660 nm), and these data were analyzed by SEDFIT software (*Schuck, 2000*) and plotted with Prism 8.0.0.

## GST pull-down assay

First, to reduce the non-specific protein binding, the Glutathione Sepharose 4B affinity beads (GST beads) (GE Healthcare) were washed three times with buffer 25 mM HEPES pH 7.4, 150 m M KCl containing 0.1% bovine serum albumin (BSA) and incubated at 4°C overnight. Then, 4 µM GST fused proteins (GST-SN25 FL or mutants) were incubated with 5 µM different Rph3A fragments or other proteins and 20 µl 50% (v/v) GST beads to a final volume of 100 µl with buffer 25 mM HEPES pH 7.4, 150 mM KCl containing 0.1% BSA. 4 µM purified GST-free proteins were used in the control groups. After gentle shaking at 4°C for 3 hr, beads were washed three times using buffer 25 mM HEPES pH 7.4, 150 mM KCl with 500×$g$, 10 min. The bound proteins were eluted with 32 µl buffer 20 mM Tris pH 8.0, 150 mM NaCl containing 20 mM L-glutathione. Samples were separated by SDS-PAGE and the protein bands were detected by Coomassie blue staining. In *Figure 1G*, the bound proteins were analyzed by Western blot and probed with mouse monoclonal anti-His$_6$ (Proteintech; 66005-1-lg). Each experiment was repeated at least three times. Data were analyzed by ImageJ and Prism 8.0.0.

## Lipid sedimentation assay

The liposome contained PC 58%, PE 20%, PS 20%: PI(4,5)P2 2% was prepared as described before (*Yang et al., 2015*). The detergent was removed by PD-10 desalting column (GE Healthcare) in buffer 25 mM HEPES pH 7.4, 150 mM KCl. Increasing concentrations of SN25 (1–206) or (11–206) (5–20 µM) were incubated with 5 µM Rph3A FL in the presence of 0.4 mM liposome with 0.5 mM Ca$^{2+}$ at room temperature for 30 min. Mixtures were centrifuged with 20,000×$g$ for 1 hr at 4°C. The supernatant (S) was acquired and the precipitate (P) was resuspended with equal volume buffer 25 mM HEPES pH 7.4, 150 mM KCl. Samples were separated by SDS-PAGE and the protein bands were detected by Coomassie blue staining. Each experiment was repeated at least three times. Data were analyzed by ImageJ and Prism 8.0.0.

## DCVs docking and secretion assay in PC12 cells

For SN25 knockdown in PC12 cells, the lentiviral expression vectors (control L309 plasmids, L309-SN25 shRNA plasmids, L309-SN25 FL or L309-SN25 11–206 rescue plasmids) were individually co-transfected with three helper plasmids (pRSVREV, pMDLg/pRRE, and pVSVG) into HEK293T cells as described before (*Wang et al., 2020*). The lentiviruses were harvested after 48 hr and concentrated by sucrose density gradient centrifugation, finally resuspended with 20 µl PBS buffer.

For the DCV docking and secretion assay, PC12 cells were cultured in RPMI Medium 1640 basic (1×) (GIBCO) supplemented with 10% FBS (GIBCO) at 37°C in a 5% CO$_2$ atmosphere at constant humidity. For TIRF imaging in *Figure 3*, PC12 cells were plated on 20 mm glass bottom dish (NEST, China). Then, 1 µg of NPY-td-mOrange2 plasmids were co-transfected with 20 µl prepared SN25 knockdown or rescue lentiviruses by using LipofectAmine 3000 (Invitrogen, USA) according to the manufacturer's instructions. After 48–72 hr, cells were imaged by TIRF microscopy on a Nikon Ti inverted microscope equipped with a 100× oil-immerse objective (NA 1.49) and an EMCCD camera. To analyze the DCV docking, the imaging was performed in a basal buffer (15 mM HEPES, pH 7.4, 145 mM NaCl, 5.6 mM KCl, 2.2 mM CaCl$_2$, 0.5 mM MgCl$_2$, and 5.6 mM glucose), and the docked DCVs were monitored by NPY-td-mOrange2. Spot detector plugin of Icy software was used to analyze the number of docked DCVs. In order to monitor the number of fusion events in per cell, a common highly K$^+$ stimulation buffer (15 mM HEPES, pH 7.4, 95 mM NaCl, 60 mM KCl, 2.2 mM CaCl$_2$, 0.5 mM MgCl$_2$, and 5.6 mM glucose) was used to displace the basal buffer to depolarize PC12 cells and trigger

vesicles fusion. We monitored exocytosis of NPY-td-mOrange2 at the single-vesicle level as described before (*Zhou et al., 2019*). Exocytosis events were scored manually. Orange fluorescence was excited with a 532 nm laser, and EGFP fluorescence was excited with a 488 nm laser. Images were analyzed by NIS-Elements viewer 4.20 and Prism 8.0.0 software.

In *Figure 4*, the procedure of Rph3A knockdown in PC12 cells and TIRF imaging assay was the same as above.

## Immunoblotting

The cell culture procedure was the same as above. In *Figure 3A* and *Figure 4E*, cells were harvested at 72 hr after transfection. Then, collected cells were solubilized with 100 μl RIPA lysis buffer in 4°C for 1 hr. The insoluble components were removed by 12,000×*g*, 10 min and the supernatants were boiled with 5× SDS loading buffer for 10 min. Equal quantities of protein were subjected to 15% SDS-PAGE followed by immunoblotting with anti SN25 antibody (Proteintech; 14903-1-AP) (*Figure 3A*), anti Rph3A antibody (Proteintech; 11396-1-AP) (*Figure 4E*), and anti-β-actin antibody (Proteintech; 66009-1-Ig). Immunoreactive bands were visualized either with HRP-conjugated goat anti-mouse (Proteintech; SA00001-1) or with anti-rabbit IgG (Proteintech; SA00001-2) and detected by ECL reagent.

## FRET assay

For the SNARE complex assembly assay in solution in *Figure 5*, the protein labeling procedure was performed as described before (*Yang et al., 2015*). Purified Syb2 (29–93, S61C) was labeled with 6× molar excess BODIPY FL N-(2-aminoethyl)-maleimide (BDPY) (Molecular Probes) and SN25 FL (1–206, R59C) or truncation mutant (11–206, R59C) were labeled with 5× molar excess Tetramethylrhodamine-5-maleimide (TMR) (Molecular Probes) in 25 mM HEPES pH 7.4, 150 mM KCl buffer in 4°C for overnight. The labeling reaction was stopped by 10 mM DTT. Excess fluorescent dyes were removed by a PD-10 desalting column (GE Healthcare) in buffer 25 mM HEPES pH 7.4, 150 mM KCl. In *Figure 5B,D*, 2 μM Syx1 (2–253) and 1 μM Syb2 S61C-BDPY were incorporated into a mixture of TMR labeled SN25 FL or 11–206 (2 μM) and Rph3A WT or mutants (8 μM). FRET assays were carried out with a PTI QM40 fluorescence spectrophotometer at 20°C using a 1 cm quartz cuvette. Donor fluorescence (DiI) was monitored with excitation and emission wavelength of 485 and 513 nm, respectively.

For the trans-SNARE complex assembly in *Figure 5*, the Syx1-liposome was reconstituted with Syx1 (1–288) contained 60% POPC, 20% POPE, and 20% DOPS, and the Syb2-liposome was reconstituted with Syb2 (29–116, S61C) contained 60% POPC, 20% POPE, and 20% DOPS. Lipid mixtures were dried in glass tubes with nitrogen gas and followed by vacuum for 3 hr. Dried lipid films were resuspended in a buffer containing 25 mM HEPES pH 7.4, 150 mM KCl, 1% β-OG (Amresco, Solon, OH) and vortexed for 5 min. The protein/lipid ratio for liposomes reconstituted with Syb2 (29–116, S61C) was about 1:500, and the protein/lipid ratio for liposomes reconstituted with Syx1 (1–288) was about 1: 500 at a final concentration of 4 mM total lipids. The liposome protein mixtures were incubated at room temperature for 30 min and then removed the β-OG by PD-10 desalting column (GE Healthcare) in buffer 25 mM HEPES pH 7.4, 150 mM KCl. The liposomes reconstituted with Syb2 (29–116, S61C) were labeled with 6× molar excess BDPY (Molecular Probes), and SN25 Δ9 (1–197, R59C) or SN25 (11–206 and Δ9, R59C) were labeled with 5× molar excess TMR (Molecular Probes). In *Figure 5G, I*, 0.5 mM Syx1-liposome (the concentration of Syx1 was about 1 μM) were mixed with 0.05 mM Syb2-liposome (the concentration of Syb2-BDPY was about 0.1 μM) and in the presence of 5 μM SN25 Δ9-TMR or SN25 (11–206 and Δ9)-TMR and Rph3A WT or mutants (8 μM). The FRET assay was carried out with FluoDia T70 at 28°C. BDPY was monitored with excitation and emission wavelength of 486 and 530 nm, respectively. Each experiment was repeated at least three times. The decreased BDPY intensity was analyzed by Prism 8.0.0 software.

## Lipid mixing assay

In *Figure 5—figure supplement 1*, the liposome (acceptor) was reconstituted with Syx1 (1–288) contained 57% POPC, 20% POPE, 20% DOPS, 1% PI(4,5)P2, and 2% DiD, and the liposome (donor) was reconstituted with Syb2 (1–116) contained 58% POPC, 20% POPE, 20% DOPS, and 2% DiI. The liposomes were prepared as described above. In the reactions, 0.2 mM acceptor liposomes were mixed with 0.2 mM donor liposomes in the presence of 10 μM SN25, SN25 (11–206), or SN23 and 10 μM Rph3A WT or mutants and 2 mM EDTA. The lipid mixing assay was carried out with FluoDia T70

at 28°C. DiI was monitored with excitation and emission wavelength of 530 and 590 nm, respectively. Each experiment was repeated at least three times. The decreased DiI intensity was analyzed by Prism 8.0.0 software.

### Bimane-tryptophan quenching assay

Purified SN25 FL or 11–206 (E55W/R59C) was labeled with 10× molar excess monobrombimane (mBBr, Molecular Probes; Eugene, OR) in buffer 25 mM HEPES pH 7.4, 150 mM KCl containing 0.2 mM EDTA. After incubation at 4°C overnight, reactions were stopped by adding 10 mM DTT. Excess fluorescent dyes were removed by a PD-10 desalting column (GE Healthcare) with buffer 25 mM HEPES pH 7.4, 150 mM KCl. In this assay, 0.7 μM bimane-labeled SN25 FL or 11–206 (E55W/R59C) were mixed with 5.6 μM Rph3A or 1.4 μM Syx1 (2–253) and 1.4 μM Syb2 (29–93). Experiments were performed at 20°C in buffer A (25 mM HEPES pH 7.4, 150 mM KCl). The bimane fluorescence was excited with a 380 nm wavelength, and monitored with an emission wave scan from 400 to 600 nm on a PTI QM-40 spectrofluorometer. SN25 FL or 11–206 R59C mutation used in *Figure 6—figure supplement 2* were also labeled bimane fluorescence as above. Each experiment was repeated at least three times.

### Fluorescence anisotropy assay

Purified Syb2 (29–96, S61C) was labeled with 6× molar excess BDPY as above. In the reactions, 0.2 μM Syb2-BDPY were mixed with 2 μM Syx1 (2–253) in the presence of 2 μM SN25 WT or SN25 E55W/R59C and SN25 11–206 E55W/R59C. The anisotropy signal was detected by PTI QM40 fluorescence spectrophotometer at 37°C using a 1 cm quartz cuvette with an excitation and emission wavelength of 485 and 513 nm, respectively. Data were analyzed by Prism 8.0.0 software.

### Sequence alignment

The sequence alignment was performed using Clustal Omega.

### Statistical analysis

Prism 8.0.0 (GraphPad) and ImageJ (NIH) were used for graphing and statistical tests.

## Acknowledgements

The authors thank Josep Rizo (University of Texas Southwestern Medical Center at Dallas) for providing Rab3A Q81L construct. The authors thank Andreas Mayer (Department of Biochemistry, University of Lausanne) for providing pEXP5-NT/TOPO vector. The authors thank Zhengxing Wu (Huazhong University of Science and Technology) for sharing the EMCCD camera.

## Additional information

### Funding

| Funder | Grant reference number | Author |
| --- | --- | --- |
| National Science and Technology Major Project | 2021ZD0202501 | Cong Ma |
| National Natural Science Foundation of China | 32225024 | Cong Ma |
| National Natural Science Foundation of China | 31670846 | Cong Ma |
| National Natural Science Foundation of China | 31721002 | Cong Ma |

The funders had no role in study design, data collection and interpretation, or the decision to submit the work for publication.

## Author contributions
Tianzhi Li, Conceptualization, Data curation, Investigation, Methodology, Writing - original draft; Qiqi Cheng, Data curation, Formal analysis, Investigation, Methodology; Shen Wang, Data curation, Investigation, Methodology; Cong Ma, Conceptualization, Supervision, Funding acquisition, Writing - original draft, Project administration, Writing - review and editing

## Author ORCIDs
Tianzhi Li (iD) http://orcid.org/0000-0003-0114-7031
Shen Wang (iD) http://orcid.org/0000-0002-5013-1039
Cong Ma (iD) http://orcid.org/0000-0002-7814-0500

## Decision letter and Author response
Decision letter https://doi.org/10.7554/eLife.79926.sa1
Author response https://doi.org/10.7554/eLife.79926.sa2

## Additional files

### Supplementary files
• MDAR checklist

### Data availability
All data generated or analysed during this study are included in the manuscript and supporting file; Source Data files have been provided.

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
