## [Editor Report]

This fundamental work is of interest to cell biologists and neuroscientists working on the molecular event underlying synaptic vesicle fusion. The authors obtain an improved and compelling description of the interaction of rabphilin with the SNARE complex protein SNAP-25. They provide a novel hypothesis of rabphilin function and how its interaction with SNAP-25 may help in the assembly of SNARE complexes.

---

## [Decision Letter]

**Decision letter after peer review:**

Thank you for submitting your article "Rabphilin 3A binds the N-peptide of SNAP-25 to promote SNARE complex assembly in exocytosis" for consideration by *eLife*. Your article has been reviewed by 3 peer reviewers, and the evaluation has been overseen by a Reviewing Editor and Suzanne Pfeffer as the Senior Editor. The reviewers have opted to remain anonymous.

Essential revisions:

1) It is necessary to verify the binding modality of Rph3A to SNAP25 in the presence of liposomes with physiological lipid composition. As the authors note, Rph3A C2B shows promiscuous binding even to the SNAP25, and thus, it is crucial to assess if the observed Rph3A C2B- N-peptide interaction is feasible and relevant in the presence of negative membranes. Assembly done in trans configuration would be more desirable, as it is more close to the situation in vivo.

2) It would be useful not to use the overexpression of rabphilin mutants and rather rely on a KD-rescue approach, as this would aid in the interpretation of the phenotype.

3) Provide some experimental/textual explanation regarding the discrepancy between these results and what is known from the KO mouse studies. In mammalian neurons rabphilin KO have almost no phenotype and even Rab3 KO has only a modest phenotype. Is this something specific for dense core vesicle fusion? A general role of rabphilin in catalyzing SNARE complex formation by chaperoning SNAP25 doesn't fit with the genetics in mice.

*Reviewer #1 (Recommendations for the authors):*

1) Considering that the Rph3A C2B also interacts with negative lipids on the plasma membrane, and this also likely involves the polybasic regions within the C2B domain, it is necessary to verify the binding modality of Rph3A to SNAP25 in the presence of liposomes with physiological lipid composition. As the authors note, Rph3A C2B shows promiscuous binding even to the SNAP25, and thus, it is crucial to assess if the observed Rph3A C2B- N-peptide interaction is feasible and relevant in the presence of negative membranes.

2) Similarly, it is necessary to study the SNARE assembly process between membranes, instead of the soluble domain only.

3) The authors need to estimate the stoichiometry of the Rph3A-SNAP25 interaction esp. since previous studies indicate that multiple Rph3A can bind SNAP25 via different interfaces. As such, it would be useful to quantify Rph3A-SNAP25 interaction using ITC/MST or similar biophysical tools to estimate the affinity and stoichiometry.

4) The PC12 analysis is carried out using the SNAP25 ∆1-11 only and there is no direct evidence that the impaired docking/fusion is due to Rph3A binding. This is a major caveat that at the minimum needs to be addressed in the text and if technically feasible with a Rph3A KD experiment.

5) One of the most interesting and novel findings is the intermolecular interplay between RBD and the C2B domains of Rph3A. Indeed, the authors describe a mutation (GLAA) that disrupts this interaction and lowers SNAP25 binding. However, there is no functional analysis of this mutation. It might be important to test this mutation in the in vitro SNARE assembly assay and if possible, in the PC12 imaging assay to check if the enhanced binding to SNAP25 is necessary for Rph3A stimulatory function.

*Reviewer #2 (Recommendations for the authors):*

I have a few suggestions that may help the authors further improve the manuscript.

1. In Figure 2., the authors concluded that the N-peptide (aa. 1-10) of SNAP-25 is essential to Rabphilin 3A binding. They aligned the sequences of SNAP-25 from different species and found that the negatively charged residues are conserved. Are these conserved residues important for SNAP-25 function?

2. In Figure 6, the authors performed bimane-tryptophan quenching assay. What's the rationale for choosing the E55W and R59C mutations? Do these point mutations affect the conformation of SNAP-25 and SNARE assembly?

3. The authors showed Rabphilin 3A accelerated SNARE assembly. How does Rabphilin 3A affect membrane fusion?

4. Is the N-peptide region found in other t-SNAREs such as SNAP-23, SNAP-47 and SNAP-29? How about Qb SNAREs in endosomal or ER-Golgi fusion?

5. Is Rabphilin 3A function in vitro specific to synaptic SNAREs? For example, does Rabphilin 3A still accelerate SNARE assembly or fusion when each SNARE is replaced with another isoform?

6. The roles of Rabs and effectors in neuronal exocytosis need further discussion. Is Rabphilin 3A specific for DCV exocytosis? How about the fusion of small synaptic vesicles? How about the Rab effector exocyst?

*Reviewer #3 (Recommendations for the authors):*

Regarding the analysis of the rabphilin mutants (Figure 4), it would be useful not to use the overexpression of rabphilin mutants and rather rely on a KD-rescue approach, as the this would aid in the interpretation of the phenotype.

Figure 5. Regarding the SNARE complex assembly analysis, this is done in solution, and an assembly done in trans configuration while technically more challenging, would be more desirable, as it is more close to the situation in vivo.

---

## [Author Response]

Essential revisions:1) It is necessary to verify the binding modality of Rph3A to SNAP25 in the presence of liposomes with physiological lipid composition. As the authors note, Rph3A C2B shows promiscuous binding even to the SNAP25, and thus, it is crucial to assess if the observed Rph3A C2B- N-peptide interaction is feasible and relevant in the presence of negative membranes.

Following the reviewers’ suggestion, we have performed the lipid sedimentation assay to test the binding of Rph3A FL and SN25 (1–206) and (11–206) in the presence of liposomes containing 58% PC, 20% PE, 20% PS and 2% PI(4,5)P2. We found that Rph3A FL bound to SN25 (1–206) but showed significantly impaired binding to SN25 (11–206) in the presence of lipid membranes. The results showed that Rph3A FL interacts with the N-peptide of SN25 in the presence of negative membranes. These data are presented in Figure 2—figure supplement 4 (pp.8, lines 170-174).

Assembly done in trans configuration would be more desirable, as it is more close to the situation in vivo.

As suggested, we performed the trans-SNARE complex assembly assay using Syx1 liposomes and Syb2 liposomes in the presence of SN25 mutant (1–206 & Δ9, deletion of the 9 residues in the C-terminal end to prevent membrane fusion (Lu, 2015)) (see new Figure 5F). Indeed, Rph3A FL accelerated trans-SNARE complex assembly, but the Rph3A K3 or GLAA mutant did not (new Figure 5G, H), consistent with the results observed in SNARE complex assembly in the absence of the membranes. However, SN25 (11–206 & Δ9) that lacks the N-peptide strongly impaired Rph3A activity in accelerating trans-SNARE complex assembly (new Figure 5I, J). Thus, these data reveal that Rph3A promotes trans-SNARE complex assembly via binding to SN25. We have added this results in the Results section (pp.12, lines 251-264).

2) It would be useful not to use the overexpression of rabphilin mutants and rather rely on a KD-rescue approach, as this would aid in the interpretation of the phenotype.

In new Figure 4, we investigated the function of Rph3A in DCV docking and fusion in PC12 cells by using knockdown-rescue approach. As expected, Rph3A KD significantly reduced DCV docking and fusion in PC12 cells, and expression of exogenous Rph3A restored the impaired DCV docking and fusion (see new Figure 4D, F-H). Rph3A-K3 mutant exhibited strongly impaired ability to restore DCV docking and fusion (see new Figure 4D, F-H). The GLAA mutant, which disrupted the intramolecular interaction between Rph3A N- and C-terminal part, also failed to restore DCV docking and fusion in PC12 cells (see new Figure 4D, F-H). These results confirmed the conclusion that the Rph3A–SN25 interaction mediated by the C_2_B bottom α-helix is essential for DCV docking and fusion in PC12 cells. We have added this new results in the Results section (pp.10-11, lines 213-230).

3) Provide some experimental/textual explanation regarding the discrepancy between these results and what is known from the KO mouse studies. In mammalian neurons rabphilin KO have almost no phenotype and even Rab3 KO has only a modest phenotype. Is this something specific for dense core vesicle fusion? A general role of rabphilin in catalyzing SNARE complex formation by chaperoning SNAP25 doesn't fit with the genetics in mice.

We have added this corresponding passage to the Discussion section:

“In contrast to the strong phenotype of Rph3A loss as observed in neuroendocrine cells, Rph3A loss has almost no phenotype in mammalian neurons (Schlüter et al., 1999). It is of note that the synapse has evolved a number of specialized sites beneath the presynaptic membrane (active zones) opposing the postsynaptic density, which are composed of multiple active-zone components required for synaptic vesicle exocytosis. Such active-zone components may take the place of Rph3A in tethering/docking vesicles at the release sites and priming them for fusion. For example, Rab-binding protein RIMs and other vesicle associated proteins, such as Munc13 and CAPS, have been found to support vesicle tethering/docking via mediating the interaction between synaptic vesicles and the plasma membrane and to support vesicle priming via promoting SNARE complex assembly (Dulubova et al., 2005; Imig et al., 2014; Quade et al., 2019; Wang et al., 2017; Wang et al., 2019; James et al., 2009). However, neuroendocrine cells lack compartmentalized active zones on the plasma membrane, in which RIMs, Munc13 and CAPS, *etc*, are dominantly distributed in the cytosol instead of the plasma membrane (Fukuda, 2004; Kabachinski et al., 2014; Kabachinski et al., 2016; Houy et al., 2022), rendering them unlikely to compensate the loss of Rph3A in DCV exocytosis. A second possibility is that other SN25 binding partners in synapses alternatively promote SN25 assembly into the SNARE complex. For example, Munc13-1 was recently found to bind the linker region of SN25 to accelerate SNARE complex assembly (Kalyana Sundaram et al., 2021). Hence, with multiple components enriched at the active zones, synapses may permit normal exocytosis in a manner independent of Rph3A.” (pp.18-19, lines 407-427)

Reviewer #1 (Recommendations for the authors):1) Considering that the Rph3A C2B also interacts with negative lipids on the plasma membrane, and this also likely involves the polybasic regions within the C2B domain, it is necessary to verify the binding modality of Rph3A to SNAP25 in the presence of liposomes with physiological lipid composition. As the authors note, Rph3A C2B shows promiscuous binding even to the SNAP25, and thus, it is crucial to assess if the observed Rph3A C2B- N-peptide interaction is feasible and relevant in the presence of negative membranes.

As mentioned in our response to Essential Revisions 1, we found that Rph3A C_2_B interacts with the N-peptide of SN25 in the presence of negative membrane (see Figure 2—figure supplement 4 and pp.8, lines 169-173).

2) Similarly, it is necessary to study the SNARE assembly process between membranes, instead of the soluble domain only.

As already elaborated in our response to Essential Revisions 1, Rph3A FL accelerated the trans-SNARE complex assembly (see new Figure 5F-J and pp.12, lines 249-262).

3) The authors need to estimate the stoichiometry of the Rph3A-SNAP25 interaction esp. since previous studies indicate that multiple Rph3A can bind SNAP25 via different interfaces. As such, it would be useful to quantify Rph3A-SNAP25 interaction using ITC/MST or similar biophysical tools to estimate the affinity and stoichiometry.

We thank the reviewer's advice. Indeed, we have tried to quantify Rph3A–SN25 interaction using ITC. Unfortunately, Rph3A FL protein was prone to aggregate under stirring in the wells in ITC instrument. Therefore, we alternatively used the GST pull-down assay to estimate the affinity and stoichiometry between Rph3A and SN25 with increasing concentrations of Rph3A FL and 6 µM GST-SN25 or GST (see Figure 1—figure supplement 2A). We used Hill equation to achieve nonlinear curve fit, where Bmax (the intensity of the bound Rph3A FL when Rph3A was added at the concentration of 10 µM) was set to 1. The dissociation constant (Kd) is 3.293 ± 0.4787 µM with a stoichiometric ratio (n) of 1.262 (see Figure 1—figure supplement 2B and pp.5, lines 95-96)

4) The PC12 analysis is carried out using the SNAP25 ∆1-11 only and there is no direct evidence that the impaired docking/fusion is due to Rph3A binding. This is a major caveat that at the minimum needs to be addressed in the text and if technically feasible with a Rph3A KD experiment.

We have followed the reviewers’ suggestion and performed Rph3A KD experiment in PC12 cells as mentioned in Essential Revisions 2 (see new Figure 4D-H and pp.10-11, lines 212-229).

5) One of the most interesting and novel findings is the intermolecular interplay between RBD and the C2B domains of Rph3A. Indeed, the authors describe a mutation (GLAA) that disrupts this interaction and lowers SNAP25 binding. However, there is no functional analysis of this mutation. It might be important to test this mutation in the in vitro SNARE assembly assay and if possible, in the PC12 imaging assay to check if the enhanced binding to SNAP25 is necessary for Rph3A stimulatory function.

To prove the effect of Rph3A-GLAA mutant, we have performed in vitro SNARE assembly assay to test the effect of GLAA mutation, and found this mutant impaired the Rph3A activity in accelerating SNARE complex assembly (see new Figure 5B-C, G-H and pp.11, lines 240-242; pp.12, lines 257-260). We have included the corresponding Rph3A-GLAA traces in new Figure 5. Similarly, Rph3A-GLAA mutant failed to restore the Rph3A knockdown phenotype in DCV docking and fusion in PC12 cells as mentioned in response to Essential Revisions 2 (see new Figure 4D-H and pp.10, lines 221-223; pp.11, lines 226-228).

Reviewer #2 (Recommendations for the authors):I have a few suggestions that may help the authors further improve the manuscript.1. In Figure 2., the authors concluded that the N-peptide (aa. 1-10) of SNAP-25 is essential to Rabphilin 3A binding. They aligned the sequences of SNAP-25 from different species and found that the negatively charged residues are conserved. Are these conserved residues important for SNAP-25 function?

Given that the acidic residues E3, D4 and D6 are conserved, we mutated these residues to alanine (A) respectively. We performed the GST pull-down assay using GST-SN25 1–82 WT or mutations, and found that the D4A mutant on SN25 N-peptide showed significantly reduced binding to Rph3A FL (see Figure 2—figure supplement 3 and pp.8, lines 166-170), suggesting that the conserved residue D4 is important for SN25 function.

2. In Figure 6, the authors performed bimane-tryptophan quenching assay. What's the rationale for choosing the E55W and R59C mutations? Do these point mutations affect the conformation of SNAP-25 and SNARE assembly?

According to the structure of SNARE complex (Sutton et al., 1998), the E55W and R59C are located on the outer surface of the bundle, which are unlikely to influence the folding of SN25 and interaction with other SNAREs. Besides, due to the bimane-tryptophan quenching effect is sensitively in short-distance electron transfer measurements (10 Å), we designed the E55W on the nearby helix of where R59C located in SN25 (Figure 6A). Anyway, we also performed the fluorescence anisotropy assay to investigate the influence of these mutants on SNARE complex assembly. Indeed, these SN25 mutants showed similar SNARE complex assembly ability as well as SN25 WT (see Figure 6—figure supplement 1 and pp.13, lines 286-287).

3. The authors showed Rabphilin 3A accelerated SNARE assembly. How does Rabphilin 3A affect membrane fusion?

To investigate the effect of Rph3A on SNARE-mediated membrane fusion, we performed lipid mixing assay as presented in Figure 5—figure supplement 1. To preclude the liposome clustering effect of Rph3A under the condition of ca^2+^, 2 mM EDTA were added in this experiment. Then, we found that Rph3A FL promoted SNARE-mediated lipid mixing in vitro, but Rph3A-K3 mutant failed to do so (see Figure 5—figure supplement 1B, C). In contrast to SN25 FL, SN25 (11–206) that lacks the N-peptide strongly impaired Rph3A activity in membrane fusion (see Figure 5—figure supplement 1D, E). The results suggest that Rph3A promotes membrane fusion dependent on Rph3A-SN25 interaction. We have added this new results in the Results section (pp.12, lines 265-270).

4. Is the N-peptide region found in other t-SNAREs such as SNAP-23, SNAP-47 and SNAP-29? How about Qb SNAREs in endosomal or ER-Golgi fusion?

Through the alignment, we observed that the N-peptide residues of SN25 are not conserved in SN23, SN47 and SN29 (see Figure 2—figure supplement 3B and pp.8, lines 164-166), and are not conserved in Qb SNAREs (Vti1A, Vti1B, GOSR1, GOSR2, Sec20) in endosomal or ER-Golgi fusion (see Figure 2—figure supplement 2C and pp.8, lines 164-166). The results suggest a specific role of SN25 in exocytosis.

5. Is Rabphilin 3A function in vitro specific to synaptic SNAREs? For example, does Rabphilin 3A still accelerate SNARE assembly or fusion when each SNARE is replaced with another isoform?

As SN23 is homologous to SN25 and involved in insulin secretion and GLUT4 exocytosis (Spurlin and Thurmond, 2006; D'Andrea-Merrins et al., 2007), we next to explore whether Rph3A regulates the SN23-mediated membrane fusion. We performed lipid mixing assay using SN23 instead of SN25, and found that the accelerating effect of Rph3A was eliminated in the presence of SN23 (see Figure 5—figure supplement 1F, G and pp.12-13, lines 270-274). Thus, the function of Rph3A in vitro is specific to synaptic SNAREs, i.e., SN25.

6. The roles of Rabs and effectors in neuronal exocytosis need further discussion. Is Rabphilin 3A specific for DCV exocytosis? How about the fusion of small synaptic vesicles?

We have discussed these questions as mentioned in Essential Revisions 3 and added this corresponding passage to the Discussion section (pp.18-19, lines 407-427).

How about the Rab effector exocyst?

In mammalian cells, exocyst is involved in membrane trafficking for cell growth and membrane protein insertion through regulating the trans-Golgi-network (TGN) derived vesicle fusion with plasma membrane (Finger and Novick, 1997; Murthy et al., 2003). However, the functional importance of exocyst in synaptic vesicle fusion has been challenged (Murthy et al., 2003). For instance, a Sec5 mutant, which destroys the exocyst complex, impairs the membrane protein addition, but synaptic transmission continues to be robust (Murthy et al., 2003).

In PC12 cells, exocyst is essential for GTP-dependent exocytosis but not for Ca^2+^-dependent DCV exocytosis, although DCVs are also derived from the TGN (Wang et al., 2004; Sadakata et al., 2013). Thus, Rph3A and exocyst may function in different pathways of exocytosis in PC12 cells and neurons.

Reviewer #3 (Recommendations for the authors):Regarding the analysis of the rabphilin mutants (Figure 4), it would be useful not to use the overexpression of rabphilin mutants and rather rely on a KD-rescue approach, as the this would aid in the interpretation of the phenotype.

We thank the reviewer's advice. We performed the Rph3A KD in PC12 cells and discussed these results in Essential Revisions 2 (see new Figure 4D-H). We have added this results in the Results section (pp.10-11, lines 213-230).

Figure 5. Regarding the SNARE complex assembly analysis, this is done in solution, and an assembly done in trans configuration while technically more challenging, would be more desirable, as it is more close to the situation in vivo.

We have performed the trans-SNARE complex assembly assay and discussed these results as elaborated in our response to Essential Revisions 1 (see new Figure 5F-J)*.* We have added this results in the Results section (pp.12, lines 249-262).